# Reverse engineering recurrent neural networks with Jacobian switching linear dynamical systems

**Jimmy T.H. Smith**
Institute for Computational and Mathematical Engineering
Stanford University
Stanford, CA 94305
jsmith14@stanford.edu

**Scott W. Linderman**
Department of Statistics
Stanford University
Stanford, CA 94305
scott.linderman@stanford.edu

**David Sussillo**
Department of Electrical Engineering
Stanford University
Stanford, CA 94305
sussillo@stanford.edu

## Abstract

Recurrent neural networks (RNNs) are powerful models for processing time-series data, but it remains challenging to understand how they function. Improving this understanding is of substantial interest to both the machine learning and neuroscience communities. The framework of reverse engineering a trained RNN by linearizing around its fixed points has provided insight, but the approach has significant challenges. These include difficulty choosing which fixed point to expand around when studying RNN dynamics and error accumulation when reconstructing the nonlinear dynamics with the linearized dynamics. We present a new model that overcomes these limitations by co-training an RNN with a novel switching linear dynamical system (SLDS) formulation. A first-order Taylor series expansion of the co-trained RNN and an auxiliary function trained to pick out the RNN's fixed points govern the SLDS dynamics. The results are a trained SLDS variant that closely approximates the RNN, an auxiliary function that can produce a fixed point for each point in state-space, and a trained nonlinear RNN whose dynamics have been regularized such that its first-order terms perform the computation, if possible. This model removes the post-training fixed point optimization and allows us to unambiguously study the learned dynamics of the SLDS at any point in state-space. It also generalizes SLDS models to continuous manifolds of switching points while sharing parameters across switches. We validate the utility of the model on two synthetic tasks relevant to previous work reverse engineering RNNs. We then show that our model can be used as a drop-in in more complex architectures, such as LFADS, and apply this LFADS hybrid to analyze single-trial spiking activity from the motor system of a non-human primate.

## 1 Introduction

Recurrent neural networks (RNNs) are a powerful and popular tool for modeling complex sequence data. They learn to transform input sequences into output sequences by using an internal state that allows data from the past to influence the current state. RNNs have been utilized in various applications such as speech recognition, sentiment analysis, music, and video [1–4]. In neuroscience, RNNs have been used for modeling large-scale neural recordings and as a generator of scientific

35th Conference on Neural Information Processing Systems (NeurIPS 2021).

hypotheses by studying the network's learned representations [5–10]. However, RNNs are generally viewed as black boxes. While there has been progress in understanding their operation on simple tasks, rigorously understanding how they solve complex tasks remains a significant challenge.

An important line of work to improve our understanding of RNN computations uses dynamical systems theory [11–19]. In particular, Sussillo and Barak [15] proposed reverse engineering a trained RNN by using numerical optimization to find the RNN's fixed and slow points. The RNN is linearized around these points, and the resulting linear approximation dynamics are studied to draw insights into how the RNN solves the task. While there are no guarantees concerning the success of this approach, empirically, linearization around fixed and slow points has led to insights in numerous applications [10, 20–26]. There are a few drawbacks of this method. First, it requires a separate numerical optimization routine after training the network. Second, it can be ambiguous which fixed point to linearize around for any given point in state space. The standard fixed point finding numerical optimization routine provides a collection of fixed points but no direct link between these points and locations in state space. Finally, simulating nonlinear RNNs with linearizations around fixed points can slowly accumulate significant error, forcing previous attempts to resort to one-step ahead dynamics generation [21]. These problems can lead to uncertainty of how well switching between linearizations around fixed and slow points describes the nonlinear dynamics.

Here, we combine ideas from reverse engineering RNNs and switching linear dynamical systems (SLDS) [27–31] to address these challenges. Given an RNN we would like to train and analyze, we introduce a separate network consisting of a novel SLDS formulation based on the first-order Taylor series expansion of the RNN equation and a learnable auxiliary function that produces the RNN's fixed/slow points. We then define a loss function that includes regularization terms to force the SLDS to approximate the RNN and switch about the RNN's fixed/slow points. After co-training these three functions with standard RNN training methods, the result is an accurate switching linear approximation of the nonlinear RNN and a trained auxiliary function that provides fixed and slow points of the RNN. This architecture and training procedure:

1. Eliminates the need for post-training fixed point finding.
2. Generalizes SLDS to be able to switch about continuous manifolds of fixed points.
3. Enables parameter sharing between the SLDS switches.
4. Allows the nonlinear RNN dynamics to be approximated by switching between local linearizations around fixed points, if possible.

The combination of these benefits significantly simplifies the process of reverse engineering an RNN using fixed points. We illustrate the method on two synthetic tasks and a neural dataset.

## 2 Review of reverse engineering RNNs and SLDSs

### 2.1 Reverse engineering RNNs with fixed points

The motivation for reverse engineering RNNs with fixed point analysis is the hypothesis that trained RNNs use mechanisms to solve tasks that are described well by the linearized dynamics around its fixed and slow points [15]. These points are state vectors $\mathbf{h}^* \in \mathbb{R}^D$ that, given an input $\mathbf{u}^* \in \mathbb{R}^U$, do not significantly change when applying the RNN update function, $\mathbf{F} : \mathbb{R}^D \times \mathbb{R}^U \to \mathbb{R}^D$. That is, $\mathbf{h}^* \approx \mathbf{F}(\mathbf{h}^*, \mathbf{u}^*; \theta)$. We can find these points numerically by minimizing a loss function

$$\mathcal{L}(\mathbf{h}) = \|\mathbf{h} - \mathbf{F}(\mathbf{h}, \mathbf{u}^*; \theta)\|_2^2, \tag{1}$$

using auto-differentiation methods [32]. In practice, one typically initializes the candidate fixed points with the hidden states produced from running forward passes given trial inputs. Once we have found the fixed/slow points, we linearize the system around these points and analyze the dynamics of the linearized system to determine how the system computes. This reverse engineering method is supported in theory by the Hartman-Grobman theorem [33–35], which says that the dynamics of a nonlinear system in a domain near a hyperbolic fixed point is qualitatively the same as the dynamics of its linearization near this point. A potential theoretical issue arises with non-hyperbolic fixed points, though empirically, this has not seemed to pose a major issue for this approach. Finally, we note a recent alternative method to finding fixed points presented in [36] that makes use of mathematical objects called directional fibers.

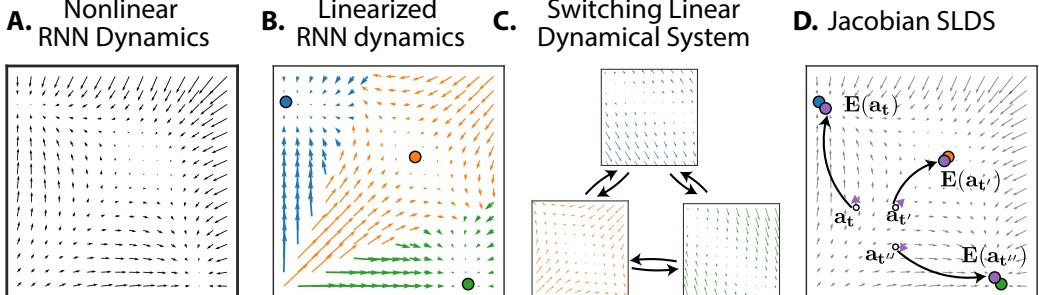

**A.** Nonlinear RNN Dynamics    **B.** Linearized RNN dynamics    **C.** Switching Linear Dynamical System    **D.** Jacobian SLDS

Figure 1: **A**. Example of a nonlinear dynamical system learned by a recurrent neural network (RNN). **B**. Linearization of the RNN dynamics around the nearest fixed point (dots). **C**. A switching linear dynamical system (SLDS) approximates the nonlinear dynamics by stochastically jumping between three linear regimes. **D**. The Jacobian SLDS co-trains an RNN (gray arrows) with an *expansion network* ($\mathbf{E}(\cdot)$), which maps the current JSLDS state (white dots, $\mathbf{a_t}$) to an expansion point (purple dots, $\mathbf{E}(\mathbf{a_t})$) near to a true fixed point of the RNN. The JSLDS linearizes the RNN dynamics around the expansion point to obtain a linear system (purple arrows) that approximates the local dynamics.

## 2.2 Switching linear dynamical systems

Models based on a linear dynamical system (LDS) are often used to model multi-dimensional time series and lend themselves well to dynamical systems analyses. The basic LDS models time series data using a latent representation that follows linear dynamics. A switching LDS (SLDS) augments the basic model with discrete states that correspond to different linear dynamics. This allows a SLDS to break down complex, nonlinear time series into a sequence of simpler local linear dynamics. Explicitly, let $\mathbf{y}_t \in \mathbb{R}^N$ denote the observation data at time $t$ and let $\mathbf{h}_t \in \mathbb{R}^D$ represent the corresponding continuous latent state. A SLDS models the expected observation value as $\mathbb{E}[y_t] = \mathbf{g}(\mathbf{h}_t)$ where $\mathbf{g}$ is a mapping from $\mathbb{R}^D$ to $\mathbb{R}^N$. Given the discrete latent state $z_t \in \{1, ..., K\}$ and input $\mathbf{u}_t \in \mathbb{R}^U$, the SLDS continuous latent states follow linear dynamics,

$$\mathbf{h}_t \sim \mathcal{N}\big(A^{(z_t)}\mathbf{h}_{t-1} + V^{(z_t)}\mathbf{u}_t + b^{(z_t)}, Q^{(z_t)}\big). \tag{2}$$

The current discrete state determines the current linear dynamics $A^{(z_t)} \in \mathbb{R}^{DxD}$, input matrix $V^{(z_t)} \in \mathbb{R}^{DxU}$, bias term $b^{(z_t)} \in \mathbb{R}^D$ and noise covariance $Q^{(z_t)} \in \mathbb{R}^{DxD}$. In the basic SLDS, a Markov transition matrix generally defines the discrete state switching probability. An extension to the SLDS model is the recurrent switching linear dynamical system (RSLDS) which allows the discrete state transition to depend on the previous continuous latent state [37–40]. This recurrent connection allows more expressivity in the model and corresponds to the idea that the current discrete state (and therefore the current dynamical regime) should depend on the current state space location.

SLDS models can often offer a balance between interpretability and expressivity for many problems. For example, one can improve the expressivity by increasing the number of discrete states, but this can come at the cost of interpretability and an increase in learnable parameters. In addition, SLDS generally requires hyperparameter tuning to determine the optimal number of discrete states to use.

## 3 Jacobian Switching Linear Dynamical System

We now present the Jacobian Switching Linear Dynamical System (JSLDS) model and training procedure. It combines ideas from reverse engineering RNNs and SLDS to achieve automated fixed point finding and accurate SLDS approximations of nonlinear RNNs.

### 3.1 Motivation

Let $\mathbf{F}$ denote a nonlinear RNN with previous state $\mathbf{h}_{t-1} \in \mathbb{R}^D$, input $\mathbf{u}_t \in \mathbb{R}^U$, and parameters $\theta$. Writing the RNN update equation and its first order Taylor series expansion around points $\mathbf{h}^*$ and $\mathbf{u}^*$

we have

$$\mathbf{h}_t = \mathbf{F}(\mathbf{h}_{t-1}, \mathbf{u}_t; \theta) \approx \mathbf{F}(\mathbf{h}^*, \mathbf{u}^*; \theta) + \frac{\partial \mathbf{F}}{\partial \mathbf{h}}(\mathbf{h}^*, \mathbf{u}^*; \theta)(\mathbf{h}_{t-1} - \mathbf{h}^*) + \frac{\partial \mathbf{F}}{\partial \mathbf{u}}(\mathbf{h}^*, \mathbf{u}^*; \theta)(\mathbf{u}_t - \mathbf{u}^*).$$
(3)

Here, $\frac{\partial \mathbf{F}}{\partial \mathbf{h}}(\mathbf{h}^*, \mathbf{u}^*; \theta) \in \mathbb{R}^{DxD}$ is the **recurrent Jacobian** that determines the recurrent local dynamics and $\frac{\partial \mathbf{F}}{\partial \mathbf{u}}(\mathbf{h}^*, \mathbf{u}^*; \theta) \in \mathbb{R}^{DxU}$ is the **input Jacobian** that determines the system's input sensitivity. If $\mathbf{h}^*$ is a fixed point of $\mathbf{F}$ for a given $\mathbf{u}^*$, then $\mathbf{F}(\mathbf{h}^*, \mathbf{u}^*; \theta) = \mathbf{h}^*$ and equation (3) yields

$$\mathbf{h}_t - \mathbf{h}^* \approx \frac{\partial \mathbf{F}}{\partial \mathbf{h}}(\mathbf{h}^*, \mathbf{u}^*; \theta)(\mathbf{h}_{t-1} - \mathbf{h}^*) + \frac{\partial \mathbf{F}}{\partial \mathbf{u}}(\mathbf{h}^*, \mathbf{u}^*; \theta)(\mathbf{u}_t - \mathbf{u}^*).$$
(4)

Eq. (4) gives a LDS that locally approximates $\mathbf{F}$ around $\mathbf{h}^*$. In principle, if one knew how to select the correct fixed point and there was always a fixed point nearby, one could use eq. (4) to run a very accurate SLDS approximation of $\mathbf{F}$ by switching between fixed points as needed. Notice the fixed point $\mathbf{h}^*$ indexes the recurrent Jacobian and input Jacobian in a manner analogous to how the discrete state $z_t$ indexes the matrices $A^{(z_t)}$ and $V^{(z_t)}$ in the SLDS of eq. (2). In practice, selecting the correct fixed point in real-time is difficult, as described above, which leads to the approach presented in the next section.

## 3.2 The JSLDS model

Our approach is to co-train the RNN, $\mathbf{F}$, with a novel SLDS formulation based on the Jacobian of $\mathbf{F}$ in the spirit of equation (4). Specifically, we introduce a separate SLDS with its own hidden state $\mathbf{a}_t \in \mathbb{R}^D$ that switches around an expansion point $\mathbf{e}_t^* \in \mathbb{R}^D$

$$\mathbf{a}_t - \mathbf{e}_t^* = \frac{\partial \mathbf{F}}{\partial \mathbf{h}}(\mathbf{e}_t^*, \mathbf{u}^*; \theta)(\mathbf{a}_{t-1} - \mathbf{e}_t^*) + \frac{\partial \mathbf{F}}{\partial \mathbf{u}}(\mathbf{e}_t^*, \mathbf{u}^*; \theta)(\mathbf{u}_t - \mathbf{u}^*).$$
(5)

Note that eq. (5) shares its parameters $\theta$ with $\mathbf{F}$ (eq. 3), i.e. given $\mathbf{e}_t^*$ and $\mathbf{u}^*$, the nonlinear RNN's parameters $\theta$ determine the update matrices. We will generally take $\mathbf{u}^*$ to be either zero (the average value for examples) or the value of a context-dependent static input. The goal is for $\mathbf{e}_t^*$ to approximate the RNN's fixed and slow points. To accomplish this, we supplement the SLDS with a nonlinear auxiliary function $\mathbf{E}$ (the **expansion network**) with separate learned parameters $\phi$

$$\mathbf{e}_t^* = \mathbf{E}(\mathbf{a}_{t-1}; \phi).$$
(6)

The expansion network $\mathbf{E}$ returns the learned expansion points and is co-trained with the nonlinear RNN and the JSLDS. Once trained, the goal is for this function to approximate the RNN's fixed/slow points. For the experiments presented in this paper, we define $\mathbf{E}$ as a 2-layer multilayer perceptron (MLP) with the same dimension per layer as the state dimension of the RNN. We discuss other potential formulations for this network in Section A.2 in the Appendix.

Eqs. (5-6) define the JSLDS model, which can be run forward in time independent of the original nonlinear RNN after training. Figure 1 illustrates the general idea. Given the previous state, $\mathbf{a}_{t-1}$, the expansion network, $\mathbf{E}$, uses this state to select the next expansion point, $\mathbf{e}_t^*$ (eq. 6). The system then updates the state by using $\mathbf{e}_t^*$ to compute the recurrent Jacobian and input Jacobian of the original nonlinear RNN $\mathbf{F}$ (eq. 5). Assuming the expansion network has learned to find fixed/slow points, switching between the points $\mathbf{e}_t^*$ corresponds to reverse engineering nonlinear RNNs using fixed points. The dependence of the expansion point $\mathbf{e}_t^*$ on the previous state of the network $\mathbf{a}_{t-1}$ links to the recurrent connection in RSLDS.

Note that eq. (5) is intended to closely follow the dynamics of $\mathbf{F}$, and we will enforce this in the training procedure. We will refer to the system comprised of eqs. (5-6) as the JSLDS. We will refer to the combination of the JSLDS and the co-trained RNN as the JSLDS-RNN system. While a limitation of our method is that it does not currently lend itself easily to a stochastic formulation like the standard SLDS, it does allow for a potential continuum of different switches using a constant number of parameters. It also automatically determines the number of switches required to solve the task instead of the hyperparameter tuning required to determine this in SLDS.

## 3.3 JSLDS co-training Procedure

We co-train together the nonlinear RNN (eq. 3) and the JSLDS (eqs. 5-6). Each network can be run forward and solve the task independently. We pass each of their states through the same

output activation function to compute two loss functions, $\mathcal{L}_{\mathsf{RNN}}$ and $\mathcal{L}_{\mathsf{JSLDS}}$, for the RNN and JSLDS, respectively. In addition, the expansion points should approximate fixed points of $\mathbf{F}$ in order to achieve a good JSLDS approximation of the RNN. We also need to ensure the JSLDS states $\mathbf{a}_t$ approximate the RNN states $\mathbf{h}_t$. We achieve these goals by adding to the total loss function a fixed point regularizer $R_e$ and an approximation regularizer $R_a$ defined as

$$R_e(\theta, \phi) = \sum_t \|\mathbf{e}_t^* - \mathbf{F}(\mathbf{e}_t^*, \mathbf{u}^*; \theta)\|_2^2 \tag{7}$$

$$R_a(\theta, \phi) = \sum_t \|\mathbf{a}_t - \mathbf{h}_t\|_2^2. \tag{8}$$

Now we define the total training loss as

$$\mathcal{L}(\theta, \phi) = \lambda_{\mathsf{RNN}} \mathcal{L}_{\mathsf{RNN}}(\theta) + \lambda_{\mathsf{JSLDS}} \mathcal{L}_{\mathsf{JSLDS}}(\theta, \phi) + \lambda_e R_e(\theta, \phi) + \lambda_a R_a(\theta, \phi) \tag{9}$$

where $\lambda_{\mathsf{RNN}}, \lambda_{\mathsf{JSLDS}}, \lambda_e$ and $\lambda_a$ control the strengths of the RNN loss, the JSLDS loss, the fixed point regularizer and the approximation regularizer, respectively. In practice, we have found these hyperparameters straightforward to select (see Section A.1 in the Appendix for a more detailed discussion). For a particular optimization iteration, we compute the loss function in eq. 9 and then update all of the parameters $\theta$ and $\phi$ at once using standard backpropagation through time (BPTT) methods for RNNs [41]. Assuming the optimization goes well, the result will be two independent trained systems, with the JSLDS approximating the nonlinear RNN to first order.

In related work, [42] introduced regularization terms that force part of the subspace of piecewise linear RNNs [43] towards plane attractors to mitigate the exploding/vanishing gradient problem [44, 45] within a simple RNN architecture. In another relevant work, [46] proposed learning interpretable nonlinear SDEs by modeling the dynamics function as a Gaussian process conditioned on the learned locations of fixed points and associated local Jacobians. We also note that our co-training procedure shares some similarities to the adversarial training in GANs [47]. However, we stress that our method shares $\theta$ between the co-trained networks, and $\theta$ and $\phi$ are each updated at the same time, i.e., we do not alternate between updating $\theta$ holding $\phi$ constant and vice versa.

## 4   Results

We analyze the JSLDS-RNN system on three examples: a synthetic 3-bit memory task, a synthetic context-dependent integration task, and multineuronal population recordings from a monkey performing a reaching task.[1] For the synthetic tasks, we use a relative error metric as in [21] to compare the quality of linearized approximation provided by JSLDS and the standard method of linearizing around the fixed/slow points (found numerically) of a standard trained RNN (without JSLDS co-training). The metric computes the relative error, $\|\mathbf{h}_t^{\mathsf{RNN}} - \mathbf{h}_t^{\mathsf{lin}}\|_2 / \|\mathbf{h}_t^{\mathsf{RNN}}\|_2$, between the nonlinear RNN state, $\mathbf{h}_t^{\mathsf{RNN}}$, and the state approximated from the linearization method, $\mathbf{h}_t^{\mathsf{lin}}$. Note that for the standard linearization method we had to resort to only computing one-step ahead dynamics predictions. This approach was necessary because running the linearized dynamics forward for many timesteps accumulates substantial error and causes the trajectory to diverge. In contrast, one-step ahead predictions were not necessary for the JSLDS.

Concretely, for the standard linearization method: we trained an RNN, numerically found its fixed/slow points, and then computed $\mathbf{h}_t^{\mathsf{lin}}$ for all of the timesteps of the held-out trials using eq. (4). To use eq. (4), we set $\mathbf{h}_{t-1}$ in that equation to $\mathbf{h}_{t-1}^{\mathsf{RNN}}$, i.e., the true previous state and we set $\mathbf{h}^*$ to be the nearest fixed/slow point (in Euclidean distance). We have to find the nearest fixed/slow point because, unlike JSLDS, the standard method does not directly link a location in state space to the fixed point one should linearize around. For the JSLDS method: we first co-trained the JSLDS and RNN together. We then simulated the JSLDS forward for the entire trajectory of each held-out trial using eqs. (5-6) to compute all of the JSLDS states $\mathbf{a}_t$. We then set $\mathbf{h}_t^{\mathsf{lin}}$ equal to $\mathbf{a}_t$ to compute the relative error for each timestep. For each method, we computed the mean relative error of all the timesteps of a held-out batch of trials. We repeated this experiment 10 times for each method by starting the training from different random weight initializations. We report the mean and standard deviation of the mean relative error across the 10 trials.

---

[1] Our implementation for the synthetic tasks is available at https://github.com/jimmysmith1919/JSLDS_public.

## 4.1 3-bit discrete memory

This task highlights how JSLDS can automatically learn to switch about a discrete number of fixed points and significantly reduces the linearized approximation error. We trained the JSLDS-RNN system to store and output three discrete binary inputs (Figure 2A) similar to the experiment described in [15]. For our purposes, the models receive three 2-dimensional input vectors where each input vector corresponds to a different channel.[2] Each input vector can take a value of $\{[1, 0], [0, 0], [0, 1]\}$ corresponding to a state of -1, 0, or +1, respectively. The models have three outputs, each of which needs to remember the last nonzero state of its corresponding input channel. The RNNs used in this experiment were GRUs [48] with a state dimension of 100 and a linear readout function. See Section A.3 of the Appendix for additional experiment details.

Projecting the co-trained RNN and JSLDS dynamics into the readout space illustrates the close agreement between the networks for predictions on held-out trials (Fig. 2B). A benefit of JSLDS is that we can use its expansion points produced along a trajectory (given the trajectory inputs) to approximate the fixed/slow points the co-trained RNN uses along the same trajectory. As a verification, we numerically found the fixed/slow points of the co-trained RNN and projected both the expansion points and the numerical fixed/slow points into the readout space (Fig. 2E). We observed that the co-trained networks learned a fixed point solution consisting of 8 marginally stable fixed points (typically 2-3 eigenvalues within .025 of (1,0) in the complex plane). This solution was robust across different random weight initializations. See Section A.3.2 of the Appendix for a detailed analysis and discussion of this solution compared to the fixed point solution found by a standard GRU without co-training (Fig. 2C). Figures 2D and F compare the mean relative error for the standard method and the JSLDS and show an example PCA trajectory of the reconstructed dynamics for each. This shows that JSLDS can simulate forward entire dynamics trajectories with much less error than the standard method (which relies on one-step ahead dynamics generation). Finally, as an additional experiment, we initialized the JSLDS co-training procedure with the trained weights of the standard GRU. We observed the fixed point solution change from the one presented in Fig. 2C to one like that presented in Fig. 2E and the same improved linearized approximations of the dynamics as presented in Fig. 2F.

## 4.2 Context-dependent Integration

This task illustrates that JSLDS can learn to switch about multiple continuous manifolds of fixed points, improve the linearized approximation of the dynamics, and be used to perform a complex analysis similar to that performed in [20]. The experiment consists of training the model to contextually decide which of two white noise input streams to integrate (Fig. 3A). The model receives two static context inputs corresponding to motion and color contexts and two time-varying white noise input streams. It is trained to output the cumulative sum of the white noise stream specified by the active context input. We used a vanilla RNN with a state dimension of 128 for the co-trained RNN and a linear readout function. See Section A.4.1 of the Appendix for more experiment details.

After co-training the JSLDS-RNN system, we observed close agreement between the JSLDS and RNN on task performance for held-out trials (Fig. 3B). Next, we analyzed the dynamics of the JSLDS for held-out trials under both contexts, set to different bias levels for both the color and motion input streams. We observed that for either context, the system integrates the relevant input using a single linear mode with an eigenvalue of 1, while the other eigenvalues decay rapidly (Fig. 3C). Next, we report the mean relative error on held-out trials for both JSLDS and linearizing a standard trained RNN in Fig. 3K. Again, JSLDS substantially improves the linearized approximation of the dynamics.

From the analysis in [20], we expect a vanilla RNN to solve this type of task by representing the integration of relevant evidence as movement along an approximate line attractor (the **choice axis**) determined by the top right eigenvector. The solution consists of two line attractors that never exist together: one exists in the motion context and the other in the color context. For a given context (and therefore a specific line attractor), the top left eigenvector (the **selection vector**) determines the amount of evidence integrated. We expect the selection vector to project strongly onto the relevant input and be approximately orthogonal to the irrelevant input. See the Mathematical Supplement Section 10 in [20] for more details.

---

[2]We reparameterized the inputs compared to [15]. This ensures the models do not have to act nonlinearly in the inputs and does not change the basic logic of the experiment since we are interested in nonlinear dynamics.

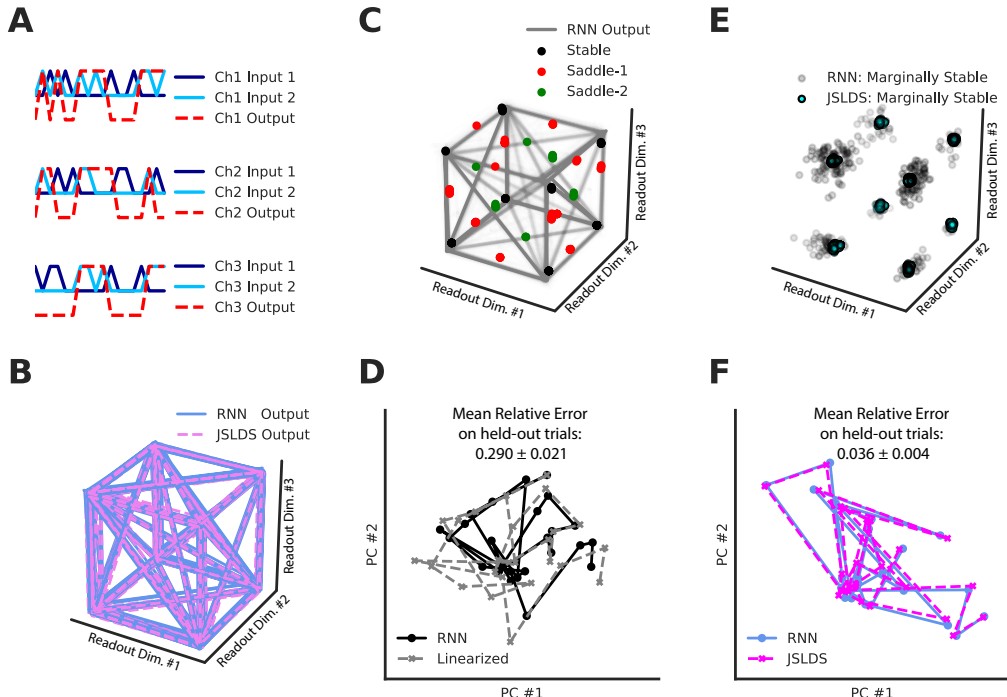

Figure 2: 3-bit memory. **A.** Two-dimensional inputs (dark blue and light blue) corresponding to input states of -1, 0, or 1 enter at random while the corresponding output (dashed red) has to remember the last non-zero input state. **B.** JSLDS closely approximates the co-trained RNN in readout space for held-out trial data. **C.** Standard GRU (no JSLDS co-training) outputs and numerical fixed points projected into readout space. **D**. Example PCA trajectory of standard GRU and linearized dynamics (one-step ahead dynamics generation) using numerically optimized fixed points. We also note the mean relative error for held-out trials. **E**. Comparison of co-trained RNN fixed points (found numerically) and JSLDS expansion points projected into readout space. The solution consists of 8 marginally stable fixed points. JSLDS has changed the fixed point solution compared to the standard GRU's solution (panel C). **F**. Example PCA trajectory of co-trained RNN and JSLDS dynamics (fully simulating dynamics forward) and the held-out mean relative error.

To verify this holds for the JSLDS, we produce a figure similar to Figures 5 and 6c from [20] by projecting the JSLDS states and expansion points into the 3-dimensional subspace meant to match the axes of choice, motion input and color input (Fig. 3D-I). Section 7.6 of the Supplementary Information of [20] describes the construction of this subspace in detail and we provide a brief description in Section A.4.2 in the Appendix. It is analogous to a regression subspace estimated from neural data in that work. It was constructed by orthogonalizing the direction of the first right eigenvectors (averaged over expansion points) and the input weight vectors corresponding to the color and motion input streams. Panels D-F and G-I correspond to the motion and color contexts, respectively. In panels D and I, we see that for the relevant context input stream, the states move along the axis of choice and the relevant input axis in proportion to the strength of the input. Panels F and G show that the strength of the nonrelevant input stream does not affect the direction of choice. Next, we analyze the global arrangement (Fig. 3J) of the motion and color context line attractors and selection vectors. As expected, we see that the selection vectors project strongly onto the relevant input axis but are approximately orthogonal to the irrelevant axis. Figure A.3 in the Appendix presents the results of performing this experiment for a standard trained vanilla RNN without the JSLDS co-training. It appears the JSLDS co-training did not dramatically change the standard trained RNN's fixed point solution for this task.

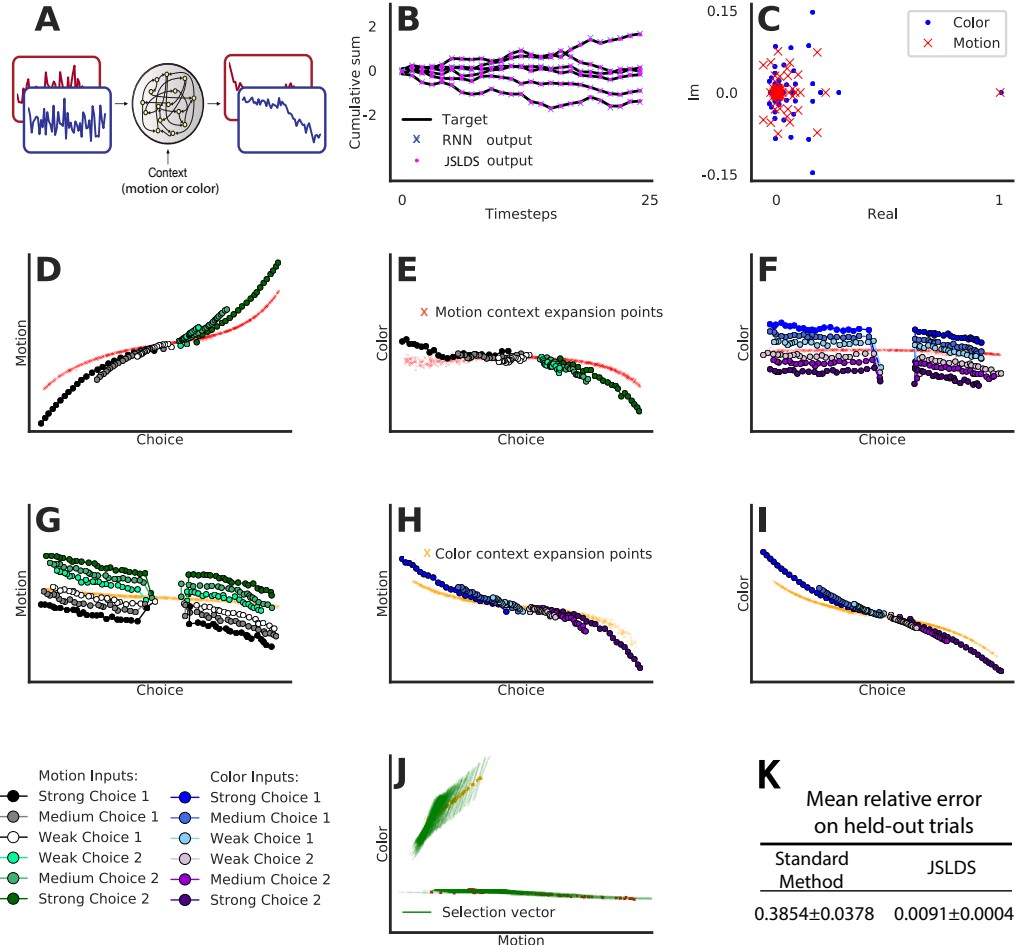

Figure 3: Context-dependent integration **A.** One of two white-noise input streams (motion or color) is selected to be integrated based on a static context input. The other stream is ignored. **B.** Sample held-out trial outputs show close agreement between JSLDS and RNN. **C.** Typical eigenvalues at a sample expansion point for motion (red x's) and color (blue dots) contexts. **D-J.** JSLDS has learned to switch between two continuous manifolds of fixed points. JSLDS states (averaged) and expansion points are projected into the subspace spanned by the axes of choice, motion and color. Movement along the choice axis represents integration of evidence and the relevant input stream deflects along the relevant input axis. The input axes of **E,F,G,H** have been intensified. The trials used in **F** and **G** are the same trials as **D-E** and **H-I**, respectively, but re-sorted and averaged according to the direction and strength of the irrelevant input. The expansion points were computed separately for motion (red x's) and color contexts (orange x's). **J.** Global arrangement of the selection vectors (green lines) and line attractor expansion points for both contexts projected onto the input axes. Inputs are selected by the selection vector (which is approximately orthogonal to the contextually irrelevant input) and integrated along the line attractor. **K.** JSLDS improves the dynamics approximation compared to linearizing a standard trained RNN.

## 4.3 Monkey reach task with LFADS-JSLDS

Finally, we illustrate how JSLDS can be dropped in as a module to improve our understanding of more complex architectures that use RNNs such as LFADS [5]. LFADS is a sequential variational auto-encoder [49, 50] used to infer latent dynamics from single-trial neural spiking data. A criticism of LFADS has been that it is hard to interpret the RNN generator that produces the dynamics. Here, we use JSLDS to improve this understanding by substituting the combined JSLDS-RNN system for

the standard GRU used in the LFADS generator (Fig. A.4 in the Appendix). We refer to this system as the LFADS-JSLDS. Once trained, we can use either the JSLDS or the co-trained RNN as the generator to produce the firing rates.

We used the monkey J single-trial maze data from Churchland et al. [51] using the same setup as Pandarinath et al. [5] to train the LFADS-JSLDS model. The data consists of 2296 trials of spiking activity recorded from 202 neurons simultaneously while a monkey made reaching movements during a maze task [52, 51] across 108 reaching conditions. We used a GRU for the RNN generator. See Section A.5.1 of the Appendix for more details. The jPCA method [51] has been applied to this data before [24, 5], so we make use of it to validate our method. It finds linear combinations of principal components that capture rotational structure in data. See [51] for the full details on jPCA.

We present the LFADS-JSLDS firing rates generated from the inferred initial condition (using the co-trained RNN generator) for several sample neurons in Fig. 4A. It was observed in [5] that the standard LFADS population dynamics on single trials exhibit rotational dynamics when projected onto the first two jPC planes. To confirm LFADS-JSLDS also exhibits this behavior, we applied jPCA to the co-trained RNN generator states (Fig. 4B). Next, the sample trials in Figure 4C show how the JSLDS generator closely approximates the RNN generator. Finally, focusing on the JSLDS generator dynamics, we learned that despite minor variations in the expansion points, the JSLDS generator eigenvalues and eigenvectors were the same at every timestep of every trial. So the JSLDS generator learned to represent the dynamics using a single, condition-independent linear system. Figure 4D displays the eigenvalues for this system.

Sussillo et al. [24] noted that the dynamics found by linearizing around the fixed point of their regularized RNN model should roughly agree with the dynamics found by applying jPCA directly to the model. Along these lines, we validate the use of JSLDS by confirming that the JSLDS generator dynamics agree with the dynamics found by fitting jPCA to the co-trained RNN generator states (Fig. 4E-F). A subspace angle analysis shows that four of the top five planes in the JSLDS state space (defined by the eigenvectors) were similar to the first four jPCA planes (Fig. 4F). Using this correspondence, we see that the eigenvalues reported by jPCA (constrained to be purely imaginary) revealed four frequencies that closely agreed with the top four frequencies found by analyzing the JSLDS generator dynamics (Fig 4E). See Section A.5.2 for more details on this subspace analysis. The correspondence between the jPCA and the JSLDS dynamics validates the use of JSLDS for the LFADS generator. Given this, we conclude that LFADS-JSLDS learns to represent the dynamics for this task using a single, condition-independent linear system, a fact that was not obvious a priori. See Section A.5.3 of the appendix for a discussion of numerically finding the fixed points of a trained LFADS model without the JSLDS co-training. We observed that the JSLDS co-training did not dramatically change the standard LFADS model's fixed point solution for this task.

## 5   Discussion

Inspired by ideas from reverse engineering RNNs [15, 20–22, 24, 25, 10] and SLDS models [37–40], this work addresses the challenging problem of improving our understanding of how RNNs perform computations. We introduced a new model, the JSLDS, to improve our ability to reverse engineer RNNs. We applied it in various settings and to different architectures: GRUs, Vanilla RNNs, and LFADS models. The JSLDS does not require post-training fixed point optimizations, significantly reduces the approximation error associated with reconstructing the nonlinear dynamics using the locally linearized solutions, and maps each point in state space to a fixed point. These benefits significantly improve our ability to reverse engineer RNNs, assuming a state-dependent SLDS can provide a good approximation of the original nonlinear RNN trained on a particular task.

Furthermore, the JSLDS could be a valuable tool to investigate the limits of the general framework of reverse engineering RNNS with fixed points. This is because we only expect the JSLDS approximation to break down if there is a system with nonlinear dynamics that are not well-described by switching between linearizations around fixed points. In addition to the above, JSLDS generalizes SLDS models to a potential continuum of switches with a constant number of parameters and automatically learns the required number of switches.

An area for refinement is the expansion network. We observed in some experiments that the expansion network might produce clusters of slightly varying expansion points that all define a single linear

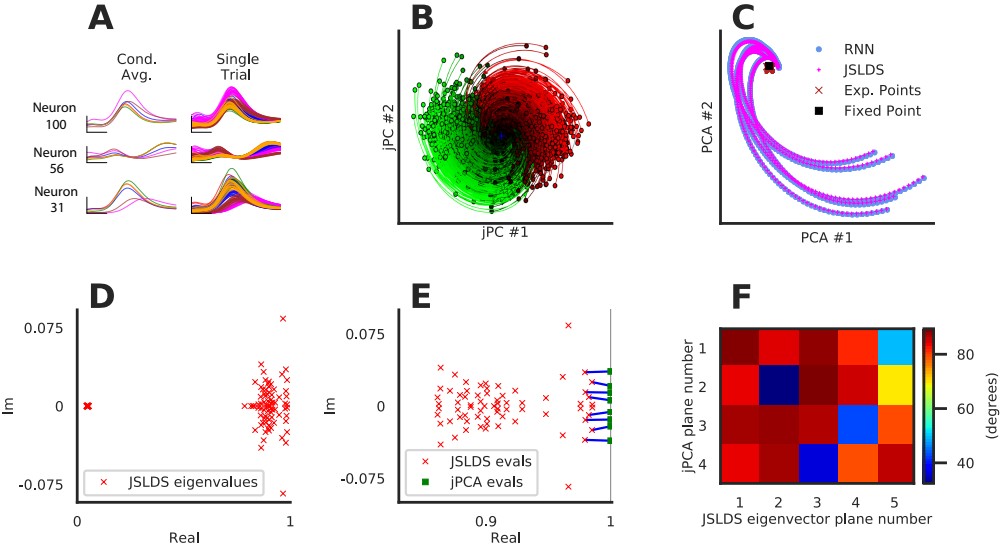

Figure 4: Monkey maze task. **A.** LFADS-JSLDS firing rates generated from inferred initial conditions for sample neurons. **B**. Projection of the co-trained RNN generator hidden states onto the first 2 jPC planes. We see the generator exhibits the expected rotational dynamics. **C**. Sample trial dynamics show low approximation error between RNN and JSLDS. Note also the JSLDS expansion points and RNN numerical fixed point. **D**. Eigenvalues of the JSLDS at a single timestep of a single trial. Our observation is that these eigenvalues are the same for every timestep of every trial, i.e. LFADS-JSLDS has learned to organize the movement dynamics with a single condition-independent linear system. **E**. Top 70 eigenvalues from the same linear system shown in D along with the purely imaginary eigenvalues associated with the jPCA analysis (green squares). The jPCA eigenvalues are connected (blue line) to their corresponding JSLDS eigenvalues given by the subspace analysis in F. **F**. Subspace analysis comparing the jPC planes and planes corresponding to the top five complex eigenvalue pairs of the JSLDS generator. Color indicates the minimum subspace angle between the corresponding planes. Angles of 30-40 degrees indicate highly overlapping subspaces.

system instead of just producing a single expansion point. Perhaps an additional loss function penalty or more specific architectures for particular tasks could help reduce this variation.

Finally, in the 3-bit memory task, the JSLDS co-training changed the fixed point structure the co-trained RNN used to solve the task compared to the standard GRU solution. We additionally observed improved linearized dynamics approximations with this new solution. These observations provide evidence that JSLDS can regularize a nonlinear RNN towards solutions better described by switching between linearized dynamics around fixed points. This regularization towards a switching linear structure could potentially have beneficial performance effects for robustness and generalization on held-out data. However, more in-depth and larger-scale studies are required to quantify these potential effects. We also note that our method could potentially suffer from the same theoretical limitation discussed for the previous reverse engineering method in Section 2.1 when near non-hyperbolic fixed points due to the Hartman-Grobman theorem. However, the potential for our method to bias the nonlinear RNN solutions towards solutions well approximated by switching between linearizations around fixed points could alleviate this concern in practice.

**Broader Impact** While understanding how RNNs perform complex computations could eventually help bound expected model behavior, identify biases, improve robustness to adversarial inputs and suggest ways to improve performance, we foresee no immediate societal consequences of this work. Overall, this work makes it easier to reverse engineer RNNs and continue to gain insight into how these models work to benefit both neuroscience and machine learning.

**Acknowledgements** We would like to thank Mark Churchland, Matt Kaufman and Krishna V. Shenoy for access to the monkey maze data.

J.T.H.S. received funding support from a Stanford Graduate Fellowhip in Science and Engineering (Mayfield fellowship). S.W.L. was supported by grants from the Simons Collaboration on the Global Brain (SCGB 697092) and the NIH BRAIN Initiative (U19NS113201 and R01NS113119). D.S. was supported by a grant from the Simons Foundation (SCGB 543049, DCS).

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
