# Supplementary Material: Reverse engineering recurrent neural networks with Jacobian switching linear dynamical systems

**James T.H. Smith**
Institute for Computational and Mathematical Engineering
Stanford University
Stanford, CA 94305
jsmith14@stanford.edu

**Scott W. Linderman**
Department of Statistics
Stanford University
Stanford, CA 94305
scott.linderman@stanford.edu

**David Sussillo**
Department of Electrical Engineering
Stanford University
Stanford, CA 94305
sussillo@stanford.edu

## A  Appendix

### A.1  JSLDS hyperparameter selection

In general, we have found the JSLDS loss function strengths to be relatively easy to select (see example settings in the specific experiment sections below). However, there are various possible configurations. The following provides a general framework for how to think about these parameters:

- $\lambda_e$ should generally be relatively large. It should be prioritized higher than the other losses or regularizers since failing to find expansion points that are good approximations of the RNN's fixed points or slow points would defeat the primary purpose of the method.

- $\lambda_a$ should be large enough to ensure a small error between the JSLDS and RNN states. However, for some tasks, one may need to balance tradeoffs between $R_a$ and the losses $\mathcal{L}_{\mathsf{RNN}}$ and $\mathcal{L}_{\mathsf{JSLDS}}$.

- For most of the experiments in this paper we set the loss strengths to $\lambda_{\mathsf{RNN}} = 1$ and $\lambda_{\mathsf{JSLDS}} = 1$. For the 3-bit memory task, we observed slightly better performance by setting $\lambda_{\mathsf{RNN}} = 3$ and $\lambda_{\mathsf{JSLDS}} = 1$. Interestingly, allowing for a slight bias towards the RNN performance on this task generally led to improved performance for both the RNN and the JSLDS. However, other variations are possible.

- For example, setting $\lambda_{\mathsf{RNN}} = 1$ and $\lambda_{\mathsf{JSLDS}} = 0$ might correspond to the goal of training a nonlinear RNN to be more interpretable by not sacrificing the goals of $R_e$ and $R_a$ for the sake of JSLDS task performance.

- In the other extreme, if one were just interested in training an SLDS, setting $\lambda_{\mathsf{RNN}} = 0$ and $\lambda_{\mathsf{JSLDS}} = 1$ could provide benefits since the JSLDS learns to share parameters across expansion points.

### A.2  Expansion network formulation

In this work we considered a specific formulation of the expansion network as $\mathbf{E}(\mathbf{a}_{t-1}; \phi)$, in which the network is a 2-layer MLP that only depends on the previous state. However other formulations

35th Conference on Neural Information Processing Systems (NeurIPS 2021).

are possible. For example, the expansion network could also depend on the previous expansion point $\mathbf{E}(\mathbf{a}_{t-1}, \mathbf{e}_t^*; \phi)$, the input $\mathbf{E}(\mathbf{a}_{t-1}, \mathbf{u}_t; \phi)$, or a combination of all of these $\mathbf{E}(\mathbf{a}_{t-1}, \mathbf{e}_t^*, \mathbf{u}_t; \phi)$. It is interesting future work to study the effects of variations such as these.

## A.3  3-bit discrete memory task

### A.3.1  Experimental details

The task here consists of three 2-dimensional input vectors where each input vector corresponds to a different channel. Each input vector can take a value of $\{[1, 0], [0, 0], [0, 1]\}$ corresponding to a state of -1, 0, or +1 respectively. The models have three outputs, each trained to remember the last nonzero state of its corresponding input channel. For example, output 2 remembers whether channel 2 was last set to state +1 or -1, but ignores the channel 1 and channel 3 inputs. When a given channel receives a nonzero input that is different from its current state, it should immediately output the new state on that timestep. For a given input vector at a given timestep, we set the probability of being in any of the three states to be equal. We set the number of timesteps $T = 25$.

We trained both methods with the Adam optimizer with default settings. For the JSLDS, we set the value of $\mathbf{u}^*$ in the JSLDS to zero. Other important hyperparameters are listed in Table 1.

Table 1: Hyperparameters used for 3-bit memory task

| Model | JSLDS-RNN | Standard RNN |
|---|---|---|
| RNN type | GRU | GRU |
| Number of RNN layers | 1 | 1 |
| Hidden state dimension | 100 | 100 |
| Batch size | 256 | 256 |
| Initial learning rate | .02 | .02 |
| L2 regularization | 0.0 | 0.0 |
| Expansion network layers | 2 | n/a |
| Expansion network units/layer | 100 | n/a |
| Expansion network activation | tanh | n/a |
| $\lambda_{\text{RNN}}$ | 3.0 | n/a |
| $\lambda_{\text{JSLDS}}$ | 1.0 | n/a |
| $\lambda_e$ | 100.0 | n/a |
| $\lambda_a$ | 10.0 | n/a |

### A.3.2  JSLDS co-training fixed point solution

The fixed point solution used by the JSLDS and co-trained GRU to solve the 3-bit memory task is significantly different from the solution used by the standard GRU (without co-training). As displayed in the main paper, the standard fixed point solution consists of stable fixed points on the corners and saddle nodes in between. In contrast, the JSLDS co-training results instead in a solution that consists of only marginally stable fixed points. We note there does seem to be some variability in the expansion network that causes the expansion points to form clusters instead of distinct points. However, this variability is relatively small in the sense that within any of the distinct clusters, all the expansion points have nearly identical linearizations. This is confirmed by checking the eigenvalues and eigenvectors for the points within each cluster. Therefore, the eight distinct clusters of marginally stable expansion points define what is essentially eight marginally stable fixed points for each of the eight possible target output states.

As we presented, the JSLDS only utilizes these eight marginally stable points. In addition, when using the numerical fixed point finding method, the slowest of the numerical fixed points of the co-trained RNN also cluster around these eight points and are also marginally stable. We can also adjust the tolerance threshold used by the numerical fixed point finding method to observe less slow points. This reveals more marginally stable fixed points in between the eight corners. These marginally stable points between the corners stand in contrast to the saddle nodes present between the corners in the standard solution. As we also noted, initializing the JSLDS co-training procedure with the trained weights of the standard GRU also leads to this same marginally stable solution. I.e., the

JSLDS co-training changes the fixed point solution from the classic solution to our new solution with marginally stable fixed points. These results suggest that perhaps one can think of the JSLDS co-training as acting to make the stable corners of the standard solution less stable and the unstable saddles of the standard solution more stable, resulting in the marginally stable solution we observe.

The co-trained JSLDS-RNN solution uses these eight marginally stable fixed points to dynamically select or ignore the inputs to update the hidden state. This is made apparent by studying the top left eigenvectors of the recurrent Jacobian (which we will refer to as the selection vectors) at each of the eight clusters and how they act upon the different possible effective inputs. Recall the input vector for each of the three channels can take a value of $\{[1, 0], [0, 0], [0, 1]\}$. Because the $[0, 0]$ input will have no effect, we can focus on just six inputs corresponding to the six one-hot input vectors that could flip one of the channel output states. We can view these six inputs for $\mathbf{u}_t$ as a $6 \times 6$ identity matrix where each column represents a different input that we are interested in. The effective input for the JSLDS update equation is $\frac{\partial \mathbf{F}}{\partial \mathbf{u}}(\mathbf{e}^*, \mathbf{u}^*; \theta)(\mathbf{u}_t - \mathbf{u}^*)$. We can use our identity matrix as the different $\mathbf{u}_t$'s, which allows us to represent the different effective inputs we are interested in as a $100 \times 6$ matrix (where 100 is the hidden state dimension used in this experiment).

We can take the dot product between the selection vectors and the different effective inputs to reveal visually intuitive patterns that clarify how the selection vectors correctly select or ignore the inputs when the system is in a particular state. We can view the top 9 left eigenvectors as a $9 \times 100$ matrix and multiply this by our $100 \times 6$ effective input matrix. Figure A.1 presents the results, and we observe that the normalized dot product between the selection vectors and the effective inputs is essentially only nonzero for the effective inputs that would cause one of the channel output states to flip.

For this task, the system should immediately update a channel output state on the same timestep it receives a nonzero input that is different from the current channel state. We can observe how the system performs this update by taking the dot product between the different possible effective inputs and the readout matrix (a size $3 \times 100$ matrix). This is because the readout matrix must use the effective input to make this update immediately. Figure A.2 presents the results. We see that the dot product is essentially only nonzero for the dimensions corresponding to the effective inputs that would cause the corresponding channel output state to flip.

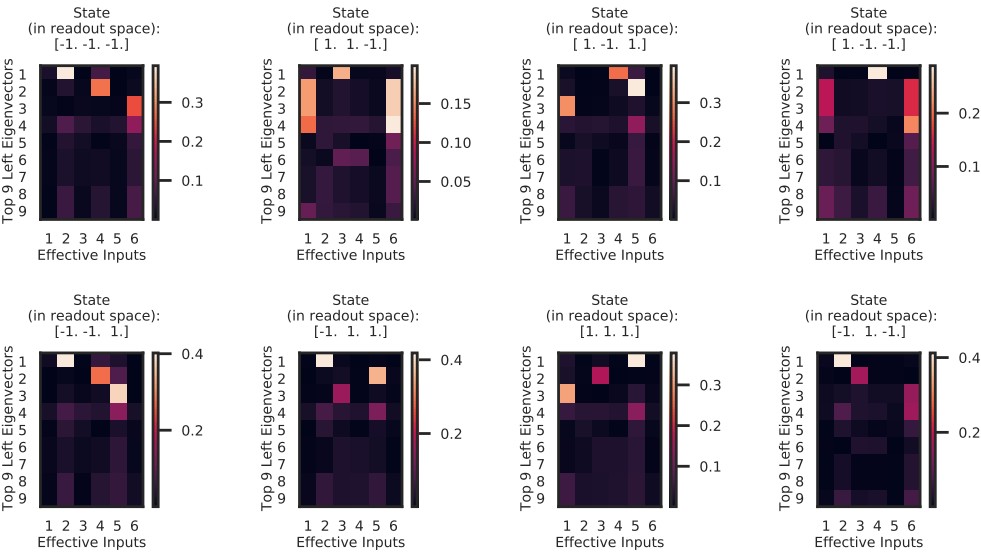

Figure A.1: Analysis of the hidden state update mechanism for the 3-bit memory task. For each of the eight expansion points the JSLDS solution uses for the eight possible output states, we take the dot product between the top nine left eigenvectors of the recurrent Jacobian $\frac{\partial \mathbf{F}}{\partial \mathbf{h}}(\mathbf{e}^*, \mathbf{u}^*; \theta)$, represented as a $9 \times 100$ matrix, and the effective input $\frac{\partial \mathbf{F}}{\partial \mathbf{u}}(\mathbf{e}^*, \mathbf{u}^*; \theta)(\mathbf{u}_t - \mathbf{u}^*)$ for each of the six one-hot inputs $\mathbf{u_t}$ we are interested in, represented as a $100 \times 6$ matrix. This results in a $9 \times 6$ matrix for each of the eight possible output states. Note that both $\frac{\partial \mathbf{F}}{\partial \mathbf{h}}(\mathbf{e}^*, \mathbf{u}^*; \theta)$ and $\frac{\partial \mathbf{F}}{\partial \mathbf{u}}(\mathbf{e}^*, \mathbf{u}^*; \theta)$ depend on the expansion point. The resulting dot product values have been normalized. We see that essentially the only nonzero results correspond to the inputs that would flip the corresponding channel state. For example, in the bottom right, state [-1,1,-1], the nonzero dot products correspond to the second, third, and sixth effective inputs. This corresponds to actual inputs of $[0, 1], [1, 0], [0, 1]$ for each of the three channels respectively. According to the task definition, these are the inputs that would cause each of the three channels to flip its respective output state. On the other hand, the first, fourth, and fifth effective inputs have no effect. This corresponds to actual inputs of $[1, 0], [0, 1], [1, 0]$ for the three channels respectively. According to the task definition, these inputs should not impact the states, as we observe.

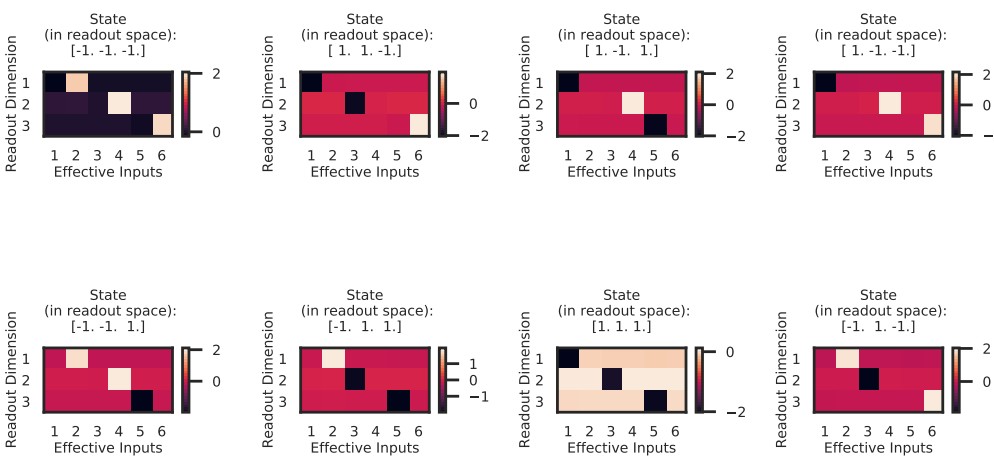

Figure A.2: Analysis of the readout state update mechanism for the 3-bit memory task. For each of the eight expansion points the JSLDS solution uses for the eight possible output states, we take the dot product between the readout matrix, represented as a $3 \times 100$ matrix, and the effective input $\frac{\partial \mathbf{F}}{\partial \mathbf{u}}(\mathbf{e}^*, \mathbf{u}^*; \theta)\,(\mathbf{u}_t - \mathbf{u}^*)$ for each of the six one-hot inputs $\mathbf{u_t}$ we are interested in, represented as a $100 \times 6$ matrix. This results in a $3 \times 6$ matrix for each of the eight possible output states. We see that essentially the only nonzero results correspond to the inputs that would flip the corresponding channel output state.

### A.4 Contextual integration task

### A.4.1 Experimental details

The experiment consists of training vanilla RNNs to contextually decide which of two white noise input streams, corresponding to motion or color contexts, to integrate. Models received two static context inputs, corresponding to motion and color contexts, and two time-varying white noise input streams of length $T = 25$. On each trial, one context input was zero and the other one, forming a one-hot encoding that indicates which input stream should be integrated. The white noise input was sampled from $\mathcal{N}(\mu, .125^2)$ at each time step, with $\mu$ sampled from $\mathcal{N}(-.01, .02^2)$ and kept static across time for each trial. The models were trained to output the cumulative sum of the correct context white noise stream at each timestep. For evaluation, the fixed $\mu$s used for the inputs were [-.04,-.02,-.009,.009,.02,.04] corresponding to strong, intermediate and weak evidence for both choices.

For $\mathbf{u}^*$ in the JSLDS, we set the dimensions that correspond to the white noise inputs to zero and set the other dimensions to the value of the context-dependent static input for each trial. We trained the system using the Adam optimizer with default settings. Other important hyperparameter settings are listed in Table 2.

Table 2: Hyperparameters used for contextual integration task

| Model | JSLDS-RNN | Standard RNN |
|---|---|---|
| RNN type | Vanilla | Vanilla |
| Number of RNN layers | 1 | 1 |
| Hidden state dimension | 128 | 128 |
| Batch size | 256 | 256 |
| Initial learning rate | .02 | .02 |
| L2 regularization | 1.0e-5 | 1.0e-5 |
| Expansion network layers | 2 | n/a |
| Expansion network units/layer | 128 | n/a |
| Expansion network activation | tanh | n/a |
| $\lambda_{\mathsf{RNN}}$ | 1.0 | n/a |
| $\lambda_{\mathsf{JSLDS}}$ | 1.0 | n/a |
| $\lambda_e$ | 100.0 | n/a |
| $\lambda_a$ | 10.0 | n/a |

### A.4.2 Subspace construction

To display the RNN trajectories in state space, we projected the JSLDS states and expansion points into the 3-dimensional subspace meant to match the axes of choice, motion input, and color input. The axis of choice for each context was determined by averaging the top right eigenvector determined by the Jacobian at each expansion point. The motion input axis was determined by the input weight vector corresponding to the motion input weight stream. Similarly, the color input axis was determined by the input weight vector corresponding to the color input stream. These three vectors were orthogonalized to create the subspace. We then projected the JSLDS states and expansion points (or RNN states and numerical fixed points for the standard trained RNN) into this subspace to create the plots.

### A.4.3 Contextual integration experiment performed without JSLDS regularization.

We repeated the contextual integration experiment with a standard trained vanilla RNN without the JSLDS co-training. After training, we numerically found its fixed points for both contexts and recreated the plot from the main paper in Figure A.3. We see that the standard trained RNN finds basically the same solution as the co-trained networks. The standard trained network eigenvalues tend to exhibit larger imaginary components, but the top eigenvalue for both contexts is still $(1, 0)$. So for the contextual integration experiment with a vanilla RNN the JSLDS does not seem to dramatically change the fixed point solution, although as observed in the main paper it still significantly improves the linearized approximation of the dynamics.

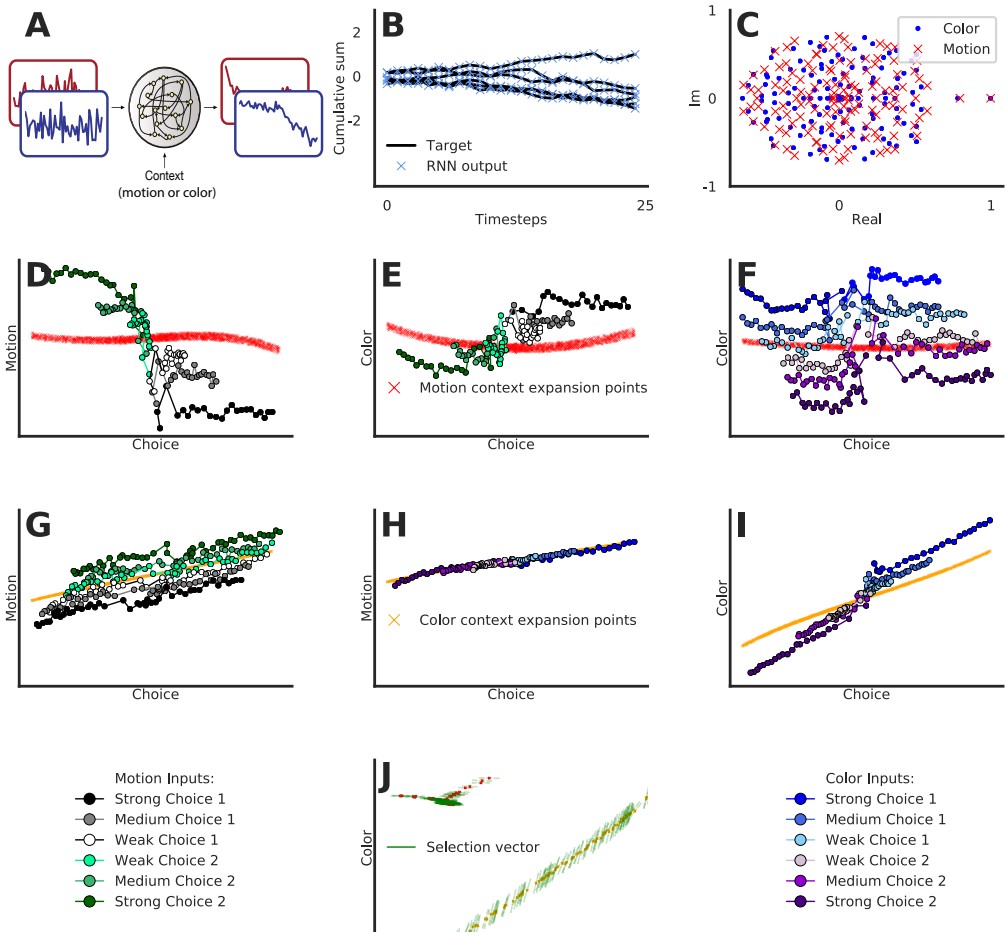

Figure A.3: Context-dependent integration for standard vanilla RNN (no JSLDS co-training) **A.** One of two white-noise input streams (motion or color) is selected to be integrated based on a static context input. The other stream is ignored. **B.** Sample held-out trial outputs and targets. **C.** Typical eigenvalues at a sample fixed point (found numerically) for motion (red x's) and color (blue dots) contexts. **D-J.** The RNN states (averaged) and fixed points are projected into the subspace spanned by the axes of choice, motion, and color. Movement along the choice axis represents integration of evidence and the relevant input stream deflects along the relevant input axis. The input axes of **E,F,G** have been intensified. The trials used in **F** and **G** are the same trials as **D-E** and **H-I**, respectively, but re-sorted and averaged according to the direction and strength of the irrelevant input. The fixed points were computed separately for motion (red x's) and color contexts (orange x's). **J.** Global arrangement of the selection vectors (green lines) and line attractor fixed points for both contexts projected onto the input axes. Inputs are selected by the selection vector (which is approximately orthogonal to the contextually irrelevant input) and integrated along the line attractor.

### A.5 Monkey reach task

### A.5.1 Experimental details

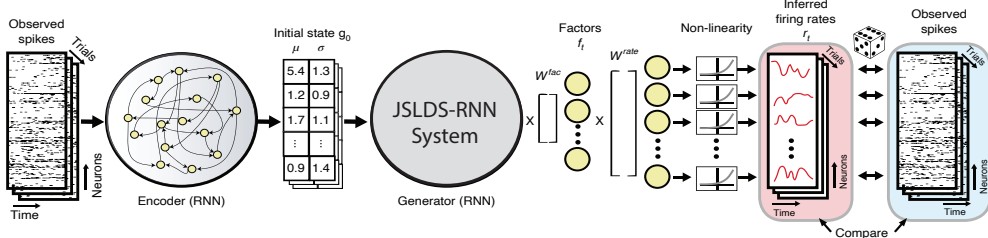

Figure A.4: The LFADS-JSLDS architecture. The JSLDS-RNN system is used as the generator. After training, the model can produce firing rates from either the JSLDS or RNN generator.

The data consists of 2296 trials of spiking activity recorded from 202 neurons simultaneously while a monkey made reaching movements during a maze task across 108 reaching conditions. The analyzed trials were 900-ms long and to train the model we used a bin size of 5 ms.

For the LFADS-JSLDS model (Fig. A.4), we used a 100 unit GRU for the RNN in the generator. To train LFADS-JSLDS, one simply includes the JSLDS loss function in the LFADS loss. We used the Adam optimizer with default settings. See Table 3 for important hyperparameter settings.

Table 3: Hyperparameters used in monkey reach task

| Model | LFADS-JSLDS | LFADS |
|---|---|---|
| RNN type | GRU | GRU |
| Generator Dimension | 100 | 100 |
| Encoder Dimension | 100 | 100 |
| Factors Dimension | 40 | 40 |
| Keep probability | .98 | .98 |
| Bin size | 5 | 5 |
| Batch size | 128 | 128 |
| Initial learning rate | .05 | .05 |
| L2 regularization | 2.0e-2 | 2.0e-2 |
| Expansion network layers | 2 | n/a |
| Expansion network units/layer | 100 | n/a |
| Expansion network activation | tanh | n/a |
| $\lambda_{\mathrm{RNN}}$ | 1.0 | n/a |
| $\lambda_{\mathrm{JSLDS}}$ | 1.0 | n/a |
| $\lambda_e$ | 100.0 | n/a |
| $\lambda_a$ | 20.0 | n/a |

### A.5.2 Subspace Analysis

We perform a subspace analysis to compare the JSLDS analysis to a jPCA analysis. The jPCA method finds linear combinations of principal components that capture rotational structure in data. Through a series of steps, it finds a transformation between a neural system at each timestep and its temporal derivative. The subspace angle refers to the angle between the planes defined by the top principal components from the jPCA analysis and the planes defined by the top JSLDS eigenvectors. To be concrete, associated with each conjugate pair of complex eigenvalues from the jPCA analysis is a conjugate pair of principal components that define a plane. Analogously, for each of the top complex pairs of eigenvalues from the JSLDS dynamics matrix, a corresponding conjugate pair of complex eigenvectors also define a plane. The subspace angle measures how similar these planes are and can be used to match up the corresponding jPCA eigenvalues (constrained to be complex) and the JSLDS eigenvalues. This is the connection displayed in Figure 4E. The fact that the top eigenvalues

of the JSLDS match up with the corresponding jPCA eigenvalues indicates our method is working correctly.

### A.5.3 LFADS experiment performed without JSLDS

We also compared the fixed point solution for the LFADS model trained without the JSLDS co-training. We trained the LFADS with the exact same hyperparameters as the LFADS-JSLDS except without the JSLDS co-training related terms. We observed this setup also learned a single linear system (Fig. A.5). So it seems in this case the JSLDS co-training did not have a significant effect on the fixed point solution.

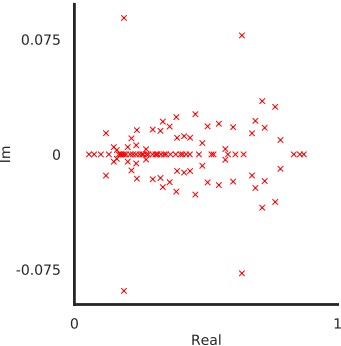

Figure A.5: Eigenvalues of the trained LFADS (without JSLDS co-training) RNN generator's Jacobian at its single fixed point.

### A.6 NeurIPS Checklist