# OpenReview forum: "Reverse engineering recurrent neural networks with Jacobian switching linear dynamical systems"
_NeurIPS.cc/2021/Conference — NeurIPS 2021 Poster_

### Official Review · Reviewer_kE2y · 2021-07-13

**Rating:** 6
**Confidence:** 5

**Summary:**

The authors introduce a framework to facilitate the analysis of recurrent neural networks (RNNs) by reverse engineering. An RNN is co-trained on a task together with a so-called Jacobian switching linear dynamical system (JSLDS). The latter is the RNN’s linearization around certain expansion points. The co-training idea results in both networks solving the task, while ensuring that the JSLDS has both similar dynamics and similar fixed points (expansion points) to the RNN.
By this construction, the trained JSLDS is easy to analyze while matching the RNN, thereby eliminating many previously needed steps of analysis. This procedure also implicitly regularizes the RNN to exhibit dynamics that can be described by the JSLDS.
The authors show the utility of the framework on three tasks previously studied for reverse engineering RNNs.


**Limitations And Societal Impact:**

The authors discuss societal impact in an appropriate manner.
Limitations: mentioned above or described by authors.


**Main Review:**

Originality: The idea of co-training a network and a linearization of it simultaneously is new. Both training and RNN and linearizing around expansion points are well-known techniques, and the authors cite previous work adequately (perhaps the work of Katz and Reggia 2017 should be mentioned as an alternative method). Perhaps GANs can be mentioned as a distant analogy.

Quality: The submission is technically sound: the authors test the new framework on a set of previously studied cases. The claims are supported by the experiments, and differences to well-studied works are described. The effect of implicit regularization on the solutions found could be better explored (detailed below).

Clarity: The submission is clearly written and well organized. It seems like it is straightforward to apply the method, but I did not try to.

Significance: There are two results in this paper. One is the introduction of a new method to analyze trained RNNs. The other is a new kind of bias or regularizer when training RNNs. The first is a welcome addition to the existing toolbox, and will probably find wide applications by many practitioners. The second is also significant, and adds to the literature on biases and the space of solutions for different tasks. This latter aspect is significant, yet underexplored in the current version.

The main issues that, in my view, could improve the paper are:
1.	The two main contributions of the paper (method and biasing) are entangled. The text currently introduces the method-result as the goal, and the biasing-result as something discovered along the way.
2.	How different are the solutions? Figure 2C,E should show this. But – they use a different color scheme, and the JSLDS in overlayed. It would be helpful to show just the RNNs using the exact same analysis. Perhaps even a 2-bit task would be better suited for this comparison. In particular, the JSLDS-RNN seems to have no saddle points between the attractors. That cannot be. There are distinct basins of attraction. This has to be properly explained.
3.	Are the solutions “simpler”, as stated by the authors. Figure 2E has a cluster of expansion points (and corresponding RNN fixed points) instead of the single attractors of Figure 2C. This seems like a more complex solution, and not a simpler one. On a similar note, the marginal nature of the clusters is also not necessarily simpler.
4.	The authors state that the method will provide RNNs that can be approximated by a linear system, if such a system exists. This raises the interesting question – which tasks lend themselves to such an approximation? Is there a negative example (otherwise the statement at L279 remains speculative)? Perhaps connected, is the choice of the authors to change the input scheme in the 3-bit example.
5.	Is it actually simpler to use the new method? Are there cases where the “original” (train, then find fixed points) method fails, and this one works? Perhaps the half-sentence at L266 “not obvious a priori” hints at this, but deserves more than half a sentence.
6.	Is there any way to limit the number of expansion points?

Specific points:

7.	The result of learning for the expansion points is not very clear. First, how does this depend on the choice of the function $G$? Second, if $F$ has multiple fixed points, will the expansion always choose the closest one?

8.	L36: proposes -> proposed
9.	Choosing the closest fixed points. L45 says it might not be the best choice. Is there an example? Furthermore, the zero, first and second order terms of the candidate fixed points might be used as a selection mechanism. If the question is the validity of a linear approximation, then these are the natural quantities.
10.	L62, feature 2. It’s not clear whether this is a good or a bad thing. It seems to complicate matters in the 3-bit example
11.	L99: limit expressivity – perhaps limit interpretability?
12.	There is no comparison of the LFADS or context integration task with “normal” RNNs. Was there a bias in the type of solutions? For readers that are not fully versed in the original work, and simply want this as an example for the new algorithm, it would be useful to compare (even in supplement).
13.	This might be a bit too much for 1 week… but: is it possible to have an intermediate algorithm. Train an RNN without other constraints. Then train the JSLDS, as a substitute for the fixed-point finder.


**Time Spent Reviewing:**

1.5

---

> ### Author Response · Authors · 2021-08-10
> **Reply to reviewer kE2y**
>
> We thank the reviewer for the positive feedback and helpful questions and comments. We are glad the reviewer appreciated the originality of our co-training idea and found the submission technically sound. We address each of the specific points below.
>
> 0. *the authors cite previous work adequately (perhaps the work of Katz and Reggia 2017 should be mentioned as an alternative method). Perhaps GANs can be mentioned as a distant analogy*: Thank you for pointing out this useful reference of Katz and Reggia 2017. We have added a reference to this paper and method in Section 2.1.  We also now point out that the co-training of the dual networks has some similarity to adversarial training in GANs.
>
> 1. *The two main contributions of the paper (method and biasing) are entangled*: We agree the effect of the implicit regularization is underexplored in this paper and is a very important and interesting future direction. We believe the significant change in fixed point solution for the 3-bit memory experiment along with the corresponding improvement in linearized approximation error is strong evidence that the JSLDS regularization has biased the solution to be better described by switching between linearized dynamics around fixed points. On the other hand, we did not observe a significant biasing effect for the other two experiments suggesting that perhaps the unconstrained training for these problems already tends to find a solution well-described by switching around fixed points. A larger study with more examples is required to fully quantify this effect.
> The main goal of the paper is to introduce the method since we think the introduction of the expansion network, the formulation of the JSLDS update equation in terms of the Jacobian of the RNN, and the co-training procedure is significant in itself even for problems in which the co-training does not significantly alter the fixed point solution.
>
> 2.
> a. *How different are the solutions? Figure 2C,E should show this. But they use a different color scheme, and the JSLDS in overlayed. It would be helpful to show just the RNNs using the exact same analysis.*: The solutions are very different. We elaborate in response to your next point. We can add a new plot in which just the two RNNs' fixed points are overlayed on top of each other as you suggest, but we are concerned this does not clarify that much. Perhaps the more detailed explanation of the JSLDS fixed point solution (as discussed below) will clarify the differences.
> b. *In particular, the JSLDS-RNN seems to have no saddle points between the attractors. That cannot be. There are distinct basins of attraction. This has to be properly explained.*: We appreciate the question and agree a much more detailed discussion of the JSLDS-RNN solution is warranted, which we now include.  There are NO saddle points in the JSLDS-RNN fixed point solution. The JSLDS-RNN solution consists of just these 8 marginally stable points. (While there is some variability in the expansion points in this experiment which causes the points to form clusters instead of single points, this "noise'' is relatively small in the sense that within a cluster, all the expansion points have nearly identical linearizations. We confirmed this by checking the eigenvalues and eigenvectors. Therefore, these 8 distinct clusters of marginally stable expansion points define what is essentially 8 marginally stable fixed points for each of the 8 possible target output states.).
> Upon further investigation, we found that these 8 points are used to dynamically select or ignore the inputs. To see this, we analyzed the left eigenvectors of the Jacobian at each expansion point. These eigenvectors act as selection vectors. When we take the dot product between these selection vectors and the effective input ($\frac{\partial F}{\partial u}(e^*, u^*; \theta)(u_t-u^*)$), it reveals visually intuitive patterns that make it clear how the selection vectors are being used to correctly select or ignore the inputs when the system is in a particular state.
> We have now elaborated on these details in Sec. 4.1. We have also included in the 3-bit memory section of the appendix useful plots and a detailed discussion that illustrate the visually intuitive patterns revealed by taking the dot product between the selection vectors and the effective inputs.
>
> 3. *Are the solutions “simpler”, as stated by the authors?*: We agree the use of "simpler" was not the most precise wording choice and have changed this to "The difference between these two solutions shows that for some tasks the JSLDS co-training procedure can change the fixed point solution. This finding, along with the improved JSLDS linearized dynamics approximations presented next provides evidence that the JSLDS can act as a regularizer towards solutions better described by switching between linear dynamics around fixed points.”
> We have also made a similar revision in the conclusion and noted the need for a larger study to quantify this.
>
> 4. *This raises the interesting question – which tasks lend themselves to such an approximation? Is there a negative example (otherwise the statement at L279 remains speculative)?*: We agree this is an interesting question. Yes, the 3-bit memory example with the standard input parameterization is a negative example in which the JSLDS procedure fails.  A linearized system as presented in equations 4 or 5 is unable to act nonlinearly in the inputs. However, assuming the ability to reparameterize inputs to overcome this, this raises the question of whether there are practical problems whose dynamics are unable to be described well by switching between linearizations around fixed points. We have revised the statements in Line 277-279 to read "The JSLDS could be a useful tool to investigate the limits of the general framework of reverse engineering RNNS with fixed points [15]. This is due to the fact that we only expect the JSLDS approximation to break down if there is a system with nonlinear dynamics that are not well-described by switching between linearizations around fixed points."
>
> 5. *Is it actually simpler to use the new method? Are there cases where the “original” (train, then find fixed points) method fails, and this one works?*:  It is a new method, so time will tell. In our hands, the answer is definitely yes. The procedure of reverse engineering fixed points of a nonlinear RNN becomes very difficult, very quickly. In particular, the JSLDS's ability to explicitly map a location in state space to the expansion point one should linearize around is highly beneficial since it allows one to know exactly which expansion point should be used at any point of a trajectory. In addition, the co-training loss function makes it such that this expansion point should be the one that minimizes the linearized approximation error. The original method does not allow one to do this precisely, since the numerical optimization just returns a collection of fixed points. Given this collection, to reconstruct the dynamics, one has to employ decision criteria to determine which fixed point should be linearized around for a given point in state space. This decision criterion adds an extra complication and uncertainty.
> In addition, the fact that reconstructing the dynamics using the previous method typically requires resorting to one-step ahead predictions (as we cite [18] in line 47 and also note for our 3-bit memory experiment in line 195) can be seen as a type of failure. The fact that the dynamics trajectories tend to diverge without this one-step ahead assist increases the uncertainty of how well the reconstructed dynamics actually describe the nonlinear solution. On the other hand, JSLDS is forced to linearize around the fixed point at each state which minimizes the linearized approximation error. This allows it to approximate a full dynamics trajectory.
>
> 6. *Is there any way to limit the number of expansion points?*: Thank you for asking this. This is an interesting question that we do not currently know the answer to. Perhaps we can formulate either a penalty or a more specific expansion network design that could reduce some of the variability observed in problems with a discrete number of fixed points such as the 3-bit memory example. The discussion of the expansion network below may be related as well.
>
> 7.
> a. *The result of learning for the expansion points is not very clear. First, how does this depend on the choice of the function $G$?*: We are assuming by $G$ you are referring to the expansion network $E$. Please let us know if we have misinterpreted this question! We found a simple 2-layer MLP was sufficient for introducing the basic model formulation in this paper. However, there are other interesting formulations of this function. This network could potentially be any nonlinear function and the inputs and outputs could also vary depending on the problem. In addition to accepting the previous state as an input, it could also accept the current problem input $u_t$ and/or the previous expansion point. The function could also potentially return the input expansion point $u^*$. We have included an extra section in the appendix discussing these other potential formulations.
> b. *Second, if $F$ has multiple fixed points, will the expansion always choose the closest one?*: We are not sure we fully understand this question. The only constraints on the expansion network learning are that it should produce fixed/slow points, the JSLDS states should match the RNN states, and the JSLDS should solve the task with low error. The expansion network should learn to do whatever is required to satisfy these constraints.

---

> > ### Author Response · Authors · 2021-08-10
> > **Reply to reviewer kE2y, continued**
> >
> > 8. *L36: proposes -> proposed*: This has been fixed!
> >
> > 9.
> > a. *Choosing the closest fixed points. L45 says it might not be the best choice. Is there an example?*: Informally, we've noticed that for example, the best expansion points on a line attractor are not the nearest. Instead, there is an angle, which all the points follow. This is a bit of "inside baseball" so we've removed the text. More broadly, the main problem with post hoc fixed point analysis of an already trained RNN is not knowing which expansion points to use.
> > b. *Furthermore, the zero, first and second order terms of the candidate fixed points might be used as a selection mechanism.*: We agree those are the natural candidates to measure, and it would be interesting to include them in future work. There are other candidates as well, such as the proximity of the expansion point to the point on the trajectory. One might also decide that the quality of the fixed point (how close is $|e^* - f(e^*)|$} to zero) is something to study.
> >
> > 10. *L62, feature 2. It’s not clear whether this is a good or a bad thing. It seems to complicate matters in the 3-bit example*: We would argue this is a good thing. The JSLDS formulation allows the system to learn to switch about continuous manifolds of fixed points such as line attractors, while also allowing it to automatically learn to switch about a finite number of fixed points (without the hyperparameter turning required for SLDS) for problems like the 3-bit memory example. As mentioned in response to your comments above, while there is some variability causing the clusters, this noise is relatively unimportant and all of the expansion points within a particular cluster define essentially the same linear system. This is all now more thoroughly explained thanks to your comment #2 above. That being said we agree it would be interesting to explore ways to limit this variability for problems with a finite number of fixed points.
> >
> > 11. *L99: limit expressivity – perhaps limit interpretability?*: We agree this sentence is currently unclear. We have revised this sentence to clarify the tradeoffs associated with increasing the number of discrete states of a SLDS (increased expressivity vs increased number of learnable parameters and decreased interpretability).
> >
> > 12. *There is no comparison of the LFADS or context integration task with “normal” RNNs. Was there a bias in the type of solutions? For readers that are not fully versed in the original work, and simply want this as an example for the new algorithm, it would be useful to compare (even in supplement)*: This is a good point. There did not appear to be a significant regularization effect for the LFADS or contextual integration experiments. We will include these comparisons in the appendix as you suggest.
> >
> > 13. *This might be a bit too much for 1 week… but: is it possible to have an intermediate algorithm. Train an RNN without other constraints. Then train the JSLDS, as a substitute for the fixed-point finder*: Thank you for the suggestion. We think there are a few different algorithm variants your comment suggests. The first is to train a normal RNN (no JSLDS regularization) and then initialize the JSLDS co-training procedure with the pretrained normal RNN weights. I.e. we train $\theta$ in the unregularized RNN and then continue to train $\theta$ and $\phi$ (the expansion network weights) in the co-training procedure. When we do this for the 3-bit memory experiment we observe that the co-training procedure changes the solution from the classic fixed point solution with saddle nodes (Fig 2C) to the JSLDS solution we observe in Fig. 2E with 8 marginally stable fixed points. We now note this in the 3-bit memory section.
> > We also tried initializing with the pre-trained unconstrained RNN weights and then just updating the expansion network weights $\phi$. This method seems to struggle, however. These initial experiments seem to suggest the interaction between updating the  $\theta$ parameters at the same time as the $\phi$ parameters is important to learn a good expansion network.  We think investigating this and other algorithms variants in more detail is interesting future work.

---

> > > ### Comment · Reviewer_kE2y · 2021-08-11
> > > **Absence of saddle points**
> > >
> > > Thanks for the detailed replied.
> > > One quick question/clarification:
> > > If I understand correctly, a single input pulse from a corner of the flip-flop brings the hidden state to another corner.
> > > This only happens for relevant inputs, and irrelevant inputs are ignored.
> > > But - what happens between these corners? Even if the dynamics do not reach these states, I would expect the "middle ground" to either contain a saddle point or be "flat" (for instance a line attractor containing the corners).
> > > I think the simplest way to check this is to interpolate the states between the two corners and see where the dynamics end from each point in this interpolation.
> > > I could be missing something here and would appreciate fully understanding this point.

---

> > > > ### Author Response · Authors · 2021-08-13
> > > > **Re: absence of saddle points**
> > > >
> > > > Thank you for the clarifying question and suggested experiment. As you suggested, we initialized the dynamics by interpolating between the corners. We observed that when dynamics trajectories start near a corner they do move in the direction toward that corner and as the initialization moves between the two corners the dynamics trajectory slowly swaps directions. However, when we run the numerical fixed point finder as a check we do not find obvious saddle nodes as we do for the "normal" RNN fixed point solution. This suggests a useful way to think about the JSLDS co-training solution is that the co-training has caused the networks to learn a solution in which the stable corners of the "normal" RNN solution become less stable and the saddles become more stable. This results in the marginally stable solution we observe.

---

### Official Review · Reviewer_DUeH · 2021-07-15

**Rating:** 6
**Confidence:** 4

**Summary:**

The central idea of this paper is to co-train a switching linear dynamical system (SLDS) together with a nonlinear RNN, such that the SLDS, due to its locally linear dynamics, can be used to gain insights into some aspects of the dynamics of the RNN. The SLDS retains the same Jacobians as the accompanying RNN, which are evaluated at approximate fixed points learned through an auxiliary function. The fixed point condition is imposed via a regularization term, as is the agreement in RNN and SLDS dynamics. The method is evaluated on 2 simple benchmarks and a neuroscience dataset.

**Ethical Concerns:**

No.

**Limitations And Societal Impact:**

Partly -- for limitations see my review above.

**Main Review:**

First, I’d like to point out that the basic problem addressed here has been solved before using piecewise linear RNNs (see for instance https://arxiv.org/pdf/1910.03471.pdf and references therein). Also the idea to impose fixed point conditions via regularization has been introduced previously in http://proceedings.mlr.press/v97/duncker19a/duncker19a.pdf (some of the ideas in this paper appear quite similar to what the authors have done). In contrast to the authors’ own formulation, these previous models enable to learn nonlinear dynamics and to explicitly compute/ determine fixed points within one and the same architecture. That is, there is no need to train two systems in parallel as in the present approach.
SLDS are also not continuous at the switching points if I recall correctly (which would make them unsuitable as a general emulator of dynamical systems).  In general, there is a wider recent ML literature on dynamical systems that is not really reviewed in here, although it seems relevant (for instance https://arxiv.org/abs/1911.00089, https://arxiv.org/abs/2009.02296, https://arxiv.org/pdf/2010.08895.pdf).

This being said, in general I still like the authors’ idea, and it may have the advantage that it’s more generally applicable. The co-trained RNN can be anything and doesn’t need to be of a particular form. But at least a proper discussion against the literature background above seems to be indicated. Although, to me the question remains, why not fit SLDSs on the data directly? Piecewise linear systems are generally expressive enough to approximate most nonlinear dynamical system (for instance Lu et al. 2017).

I was less convinced by the evaluation: No comparisons to any other method were performed (VIND, https://arxiv.org/pdf/1811.02459.pdf, https://arxiv.org/abs/1707.09049, or PLRNN as cited above, seem obvious candidates). Only single examples were presented, which could have been cherry-picked, rather than a thorough statistical evaluation based on many different benchmark simulations and parameter initializations. For instance, how would I know that some of the results like those in Fig. 2E are not completely over-interpreted?
The neuroscience examples are probably only comprehensible if one had read [5] and [17] before. Without this background, it remains largely unclear what precisely is shown in Figs. 3 and 4. For example, in Fig 4, eigenvalues of precisely what are shown here (all eigenvalues of all Jacobians?), what does jPCA do, what is a ‘subspace angle’, what exactly is meant with ‘decaying oscillatory modes’ in a linear system (stable spiral points)? Many things I could just guess as they were not explained neither in the text nor in the legend, and some of the subpanels were very unclear to me (e.g. Fig. 4E).

It was also not fully clear to me why co-training RNN and JSLDA works that much better than finding fixed points numerically posthoc and linearizing around them. In terms of runtime I guess it doesn’t make a lot of a difference whether co-training of a second system with resp. regularization terms is involved, or whether this optimization is performed afterwards? But if the correct fixed points of the original RNN can be found, why does linearizing around them work that much worse? Also in this context, can the fixed points not simply be found numerically by iterating the RNN equations forward in time from different initial conditions? That should be cheap and fast.

How is it ensured that the fixed points are actually stable? If they are repellers, trajectories would exponentially diverge from them and the SLDS approximation would break down, no?

Finally, does the RNN+SLDS learn the dynamics of the underlying experimental system in Fig. 4? Only posterior inference it seems was shown in Fig. 4A, which is not that impressive given that other models are already around that can learn the full dynamics (I conjecture that a simple LDS would have given similarly good agreements).

Bottom line: Although I’m kind of fond of the basic idea, it needs to be worked out much more rigorously and convincingly in my mind.

Minor issues:
- sect. 2.1: A fixed point is a point for which x*=F(x*) holds exactly. If it’s just approximate, it’s not a fixed point but perhaps a ‘slow point’ or ‘attractor ruin’.
- sect. 4.1: Why do the fixed points of JSLDS need to be found by running the system forward in time? This could have been done right away with the RNN? And I thought the whole point of JSLDS is that the expansion points e* are approximate fixed points and hence returned directly, let alone that they could simply be explicitly computed in linear systems?
- If there are just 8 fixed points in Fig. 2E, then what is all the jitter about? Also not sure what Fig. 2B tells me. How was the readout function g defined? There are so many unexplained details in all of the figures.
- How exactly are the motion, color, choice axes determined; are these the axes along which motion etc. cause most variance?
- What exactly is shown in Fig. 4B?
- p.9: How can a nonlinear system like the JSLDS indicate how ‘nonlinear’ a task is?

**Time Spent Reviewing:**

5

---

> ### Author Response · Authors · 2021-08-10
> **Reply to reviewer DUeH**
>
> We thank the reviewer for the very detailed and thoughtful feedback and questions. We are glad the reviewer is fond of the basic idea and recognizes the applicability to general RNN architectures.  We address each specific comment below.
>
> 1. *On Schmidt 2021 and Duncker 2019*: Thank you for pointing out these relevant references. We agree there is an important connection between our regularization approach and the use of regularization terms to affect the dynamics in these papers. We now discuss and reference this connection in Section 3.3 where we define the loss function and regularization terms.
> More generally, as we note in the introduction, the basic problem our paper focuses on is to improve upon the general framework of reverse engineering commonly used RNN architectures through linearizations around numerically computed fixed points (References 10,15,17-23 in our paper as noted in the introduction ).  The main aim of our work is to introduce a model that improves this well-established and commonly used framework.
> The Schmidt 2021 paper introduces a way to regularize piecewise linear RNNs such that part of its subspace is regularized toward plane attractors. This is a very specific RNN formulation that regularizes towards a specific fixed point solution.
> As you mentioned, our method is applicable to any RNN architecture and provides a method to analyze the dynamics for architectures that cannot be solved analytically. In addition, it does not predefine the fixed point solution that should be found and allows for the discovery of both fixed point solutions with a finite number of fixed points as well as solutions with continuous manifolds of fixed points.
> Duncker 2019 is focused on learning interpretable nonlinear SDEs by modeling the dynamics function as a Gaussian process with a certain number of fixed points. This is an interesting approach to the slightly different problem of continuous-time SDEs, as opposed to our goal of improving our understanding of general discrete-time RNNs. Nonetheless, their work is also inspired by the same reverse-engineering framework our work improves upon, and we now include a more detailed discussion of these connections. Investigating further connections between this paper and ours is interesting future work!
>
> 2. *SLDS are also not continuous at the switching points if I recall correctly (which would make them unsuitable as a general emulator of dynamical systems)*: We agree that SLDS switching points are discrete (as we note in section 2.2) and this can limit their expressivity (also noted in section 2.2, though we have improved this statement based on your comment below). One way to view our method is as a way to generalize SLDS to have continuous switching points. This allows for switching about continuous manifolds of switching points when required (as we note in line 62).
>
> 3. *On the wider recent ML literature on dynamical systems*: Thank you for both the positive feedback and the useful references. While we have referenced the relevant literature related to our primary goal of building on the established framework of reverse engineering RNNS, we agree a more broad discussion of recent literature on dynamical systems/RNNs (and where our work is situated within that) is useful.  We have now added a more detailed discussion of this in our introduction.
>
> 4. *Why not fit SLDSs on the data directly?*: This has been tried many times over many decades (e.g. see Linderman et al. 2019 for a recent example using recurrent SLDS models with neural data). The sticking points are: choosing the number of linear systems, the dynamics are not always adequate to describe the data, and there are sometimes optimization issues related to the discrete component of the optimization (i.e. the need for the training to determine which discrete state should handle which part of state space is a combinatorial optimization problem). Further, while it is true that given enough discrete switching states an SLDS should be able to approximate most nonlinear dynamical systems, the number of learnable parameters becomes large quite quickly (one has to learn a new dynamics matrix, input matrix, and bias term for each discrete state added). This is mentioned briefly at the end of Section 2.2 though we have now elaborated on these points to make it more clear.
> An advantage of the JSLDS formulation is that its update equation (provided by the Jacobian of the co-trained RNN) at any point in time is determined by the RNN's parameters ($\theta$) and the expansion point determined by the expansion network (with parameters $\phi$). This allows the JSLDS to use as many switching states as it needs while sharing a constant number of parameters across them.
> We would like to highlight however that the main goal of this work is less focused on the best way to fit a particular dataset, and more focused on introducing an idea to improve our understanding of how nonlinear RNNs perform computations, which is of high interest to both machine learning and neuroscience. We do think the ideas behind JSLDS can be useful to improve the ability to model data though and think a further investigation of this will be an important future line of work.
>
> 5. *On comparisons to other methods in the evaluation*:  Our goal in this work is to improve the well-established existing method (training a RNN, numerically finding its fixed points, and using these to reconstruct locally linearized dynamics) that is used throughout the field of work related to reverse engineering RNNs. For this reason, it seems the most relevant comparison is to compare our method to the existing method.
> Your comments raise very interesting questions in terms of how the general framework of reverse engineering RNNs using fixed points and slow points compares in terms of interpretability and improving understanding of computations compared to other approaches such as the ones you reference. However, this seems to be a larger question and is an interesting future line of work for the community. We hope our method will be a useful tool for users to explore the usefulness and limits of the general reverse engineering framework.
>
> 6. *Only single examples were presented, which could have been cherry-picked, rather than a thorough statistical evaluation based on many different benchmark simulations and parameter initializations. For instance, how would I know that some of the results like those in Fig. 2E are not completely over-interpreted?*:   In terms of the benchmark simulations being potentially cherry-picked, the 3-bit memory example and the contextual integration are canonical examples in our field, and have been used heavily in previous works on reverse engineering RNNs. They are also of high interest to the neuroscience community in general. The 3-bit memory experiment is an example that forces JSLDS to switch between a discrete number of fixed points, while the contextual integration example highlights a task in which the JSLDS needs to learn to switch about continuous manifolds of fixed points.
> As to whether or not the results have been cherry-picked within a particular experiment, for the 3-bit memory experiment, we now report the mean and standard deviation over 10 random initializations for the error associated with the linearized approximations of the dynamics (JSLDS: mean: 2.329e-4, std: 1.326e-5; Previous method: mean: 3.68e-2, std: 4.75e-3.). For visual results, we can verify that the multiple reruns from different random initializations always result in similar fixed point solutions. We also ran an additional experiment in which we trained a normal RNN (no JSLDS regularization) on the 3-bit memory task and then initialized the JSLDS/RNN co-training procedure with these pretrained weights.  We observe that the JSLDS co-training changes the fixed point solution from that observed in Fig. 2C to that observed in Fig. 2E. This provides further evidence that the JSLDS solution is not simply a result of random initializations.  We now note this result.
> Finally, Colab notebooks were also submitted in the supplemental information to allow a user to explore different initializations and hyperparameter settings.
>
> 7. *The neuroscience examples are probably only comprehensible if one had read [5] and [17] before.*: We appreciate this feedback. We chose to analyze examples from these two papers since each has been influential in the intersection of machine learning and neuroscience. However, we want to ensure the results are clear for a general audience less familiar with these works and have added more details to the experiment sections, figure captions and appendix to address this. This includes addressing your specific comments below.
>
> 8. *For example, in Fig 4, eigenvalues of precisely what are shown here (all eigenvalues of all Jacobians?)*: These are the eigenvalues for a single Jacobian at a single timestep of a single trial. But our finding is that the eigenvalues are the same for all timesteps of all trials since the system has learned a single condition-independent linear system (line 254-255). We have now made this more explicit.
>
> 9. *what does jPCA do?*: jPCA finds linear combinations of principal components that capture rotational structure in data. Through a series of steps, it finds a transformation between a neural system at each timestep and its temporal derivative.  While the full details are in [38], we have added a section in the appendix that gives an overview of the method.

---

> > ### Author Response · Authors · 2021-08-10
> > **Reply to reviewer DUeH, continued**
> >
> > 10. *what is a ‘subspace angle’?*:  Here, the subspace angle refers to the angle between the planes defined by the top principal components from jPCA and the planes defined by the top JSLDS eigenvectors. This subspace analysis terminology and the corresponding details come from [21]. To be concrete, associated with each conjugate pair of complex eigenvalues from the jPCA analysis is a conjugate pair of principal components that define a plane. Analogously, for each of the top complex pairs of eigenvalues from the JSLDS dynamics matrix, there is a corresponding conjugate pair of complex eigenvectors which also define a plane.  The subspace angle is a measure of how similar these planes are and can be used to match up the corresponding jPCA eigenvalues (constrained to be complex) and the JSLDS eigenvalues. This connection is shown in Figure E and discussed in lines 260-264. The fact that the top eigenvalues of JSLDS match up with the corresponding jPCA eigenvalues can be viewed as a validation our method is working correctly since jPCA is an established method used for many applications in computational neuroscience. A similar analysis was used to validate an experiment performed in Figure 7 of [21]. We have adjusted the wording in this section and added more details to the LFADS experiment section of the appendix to make this more clear.
> >
> > 11. *what exactly is meant with ‘decaying oscillatory modes’ in a linear system (stable spiral points)?*: This refers to the top 5 complex eigenvalue pairs. We originally used the same language as Figure 7 of [21], but we have now changed this to "Top 5 complex eigenvalue pairs" for a general audience.
> >
> > 12.  *Many things I could just guess as they were not explained neither in the text nor in the legend, and some of the subpanels were very unclear to me (e.g. Fig. 4E)*:  We appreciate this feedback and have added additional details in the corresponding sections of the appendix for readers that may not be as familiar with [5,17,21]. Specifically for Fig. 4E, we hope the correspondence between the JSLDS eigenvalues and the jPCA eigenvalues as verified by the subspace analysis in Fig. 4F is now more clear after the explanation of subspace angles discussed above.
> >
> > 13. *It was also not fully clear to me why co-training RNN and JSLDS works that much better than finding fixed points numerically posthoc and linearizing around them....But if the correct fixed points of the original RNN can be found, why does linearizing around them work that much worse?*: With either method, one should generally be able to find the fixed points. The important question is: How do we reconstruct the RNN dynamics once the fixed points are found? A major downside with the previous method is that the numerical optimization just returns the collection of fixed points. As mentioned in the introduction (lines 43-45) and the 3-bit memory section (lines 190-195), there is no direct connection to the parts of the dynamics trajectory these points correspond to, which can lead to ambiguity regarding which point one should linearize around. In short, using the previous method, things become very confusing, very quickly in many circumstances. A major advantage of JSLDS is that the expansion network provides an explicit mapping from a location in state space to the fixed point the network should linearize around. In addition, due to the co-training loss function, we expect this fixed point to be the one that minimizes the linearized approximation error.  So at any timestep along a trajectory of an example trial (i.e. at any point in state space) we know precisely which expansion point the network should linearize around to minimize the approximation error.
> > In addition to the above, we expect the JSLDS co-training to potentially regularize the RNN to be better described by switching between linearizations around fixed points. The combination of the changed fixed point solution in the 3-bit memory experiment and the improved linearized approximations by the JSLDS provides evidence that this can be the case.
> >
> > 14. *Also in this context, can the fixed points not simply be found numerically by iterating the RNN equations forward in time from different initial conditions? That should be cheap and fast.*: Just running fixed point iterations would not allow for the discovery of saddle points which can be very important. For example, the canonical solution for the 3-bit memory experiment using the previous method as presented in [15] consists of multiple saddle nodes (and our results reflect this for the previous method in Figure 2C).
> >
> > 15. *How is it ensured that the fixed points are actually stable? If they are repellers, trajectories would exponentially diverge from them and the SLDS approximation would break down, no?*:  Related to the point above regarding saddle points, we actually want to find all of the unstable fixed points if they exist. The linear approximation still holds near the unstable fixed point. While it is true that (in the absence of inputs) a repeller would push the system away from this point and the linear approximation would become less accurate, the JSLDS should learn to switch to a new fixed point when it has moved too far away from the repeller for the linearized approximation to be accurate (as enforced by the co-training loss function).
> >
> > 16. *On the experimental results presented in Figure 4*: We agree other methods could also learn the dynamics well for this data. The goal here was to replicate one of the main experiments in [5] using the co-trained RNN/JSLDS as the dynamics generator of LFADS. This allows us to easily reverse engineer the potentially complicated and hard to understand LFADS generator and draw insights into how it is solving the task. The "black box" nature of this generator has been a common criticism of the LFADS method.
> >
> > Minor issues:
> >
> > 17. *If it’s just approximate, it’s not a fixed point but perhaps a ‘slow point’ or ‘attractor ruin’.*: We have changed this to slow point.
> >
> > 18. *Why do the fixed points of JSLDS need to be found by running the system forward in time?*: The expansion network function depends on the previous hidden state, $a_{t-1}$ (location in state space) of the JSLDS. One could technically choose any point in state space and just push it through the expansion network to see which expansion point it is mapped to. However, the most relevant locations to analyze will be the ones along the dynamics trajectories for example trials.
> >
> > 19. *This could have been done right away with the RNN?*: When performing the numerical optimization for the original method it is typical to initialize the candidate fixed points with the hidden states of the example trial trajectories (we have added this detail to Section 2.1). So for the original method, after training is completed, you actually need to run the network forward for all the trials you are interested in before performing the fixed point optimizations. For the JSLDS on the other hand, after training, one can easily look at just the expansion points used for individual trials if desired.
> >
> > 20. *And I thought the whole point of JSLDS is that the expansion points are approximate fixed points and hence returned directly*: The expansion network returns the next expansion point the JSLDS should linearize around given the previous hidden state ($a_{t-1}$) of the JSLDS. A major advantage is that given a point in state space along the dynamics trajectory, we can explicitly map this point to the fixed point we should linearize around. For the previous method, we have to run some kind of decision procedure such as choosing the closest fixed point to try and decide which fixed point we should linearize around (described in lines 192-195). This adds an additional complication and uncertainty in the previous method's linearized approximation (and therefore our ability to understand how the network performed the computation).
> >
> > 21. *If there are just 8 fixed points in Fig. 2E, then what is all the jitter about?*: We now provide a more detailed explanation of the JSLDS fixed point solution in the 3-bit memory experiment section. While there is some variability in the expansion points in the 3-bit memory experiment (Fig. 2E) which causes the points to form clusters instead of single points, this "noise" is relatively small in the sense that within a cluster, all the expansion points have nearly identical linearizations. We confirmed this by looking at the eigenvalues and eigenvectors. Therefore, these 8 distinct clusters of marginally stable expansion points define what is essentially 8 marginally stable fixed points for each of the 8 possible target output states.
> > We do agree that there is perhaps room for improvement to decrease this variability. Perhaps an additional penalty term or a more specific expansion network design could address this. We now mention this in the conclusion.
> >
> > 22. *Also not sure what Fig. 2B tells me. How was the readout function g defined?*: Thank you for pointing out that we did not define the readout function in the text, this has now been added. It is just a linear function. Figure 2B is simply a verification that the JSLDS output closely approximates the RNN output (lines 176-177) and that both form a cube-like solution representing the 8 possible target states (it would indicate an issue with the method if these did not match). It is analogous to the gray lines in Figure 2C for the original method.

---

> > > ### Author Response · Authors · 2021-08-10
> > > **Reply to reviewer DUeH continued, continued**
> > >
> > > 23. *How exactly are the motion, color, choice axes determined; are these the axes along which motion etc. cause most variance?*: We describe the construction in lines 224-228.  The choice axis is represented by the top right eigenvector (which we state in line 217-218). To be concrete, we construct this axis by averaging over the top right eigenvectors of the Jacobians defined at all of the expansion points (lines 226-227). Prior to the orthogonalization, the color axis and the motion axis are just the input weight vectors corresponding to the color and motion input streams respectively (lines 226-228). The 3-d subspace is constructed by orthogonalizing these 3 vectors (mentioned in line 226-228).  This is the same construction as used in [17]. We now provide a more detailed description in the contextual integration section of the appendix to motivate this construction.
> > >
> > > 24. *What exactly is shown in Fig. 4B?*: This shows a projection of the LFADS-JSLDS generator hidden state trajectories onto the first 2 jPC planes discovered by jPCA. This is a verification that LFADS trained with the RNN-JSLDS system as the generator exhibits the rotational dynamics expected from the results in [5] in which they used a standard GRU as the generator. It would indicate an issue with the method if these trajectories did not rotate similarly to [5]. We have added additional wording to the statement in lines 250-252 as well as the caption to clarify this.
> > >
> > > 25. *How can a nonlinear system like the JSLDS indicate how ‘nonlinear’ a task is?*: Thank you for the feedback on this statement, we have improved the wording to clarify. This was motivated by a sense that systems with dynamics that cannot be described well by switching betweeen linearizations around fixed points (and thus systems in which the JSLDS approximation would be expected to break down) might be considered more "nonlinear" then systems that can be.  We feel JSLDS could be a useful tool to empirically investigate this delineation. We have revised this sentence to read "The JSLDS could be a useful tool to investigate the limits of the general framework of reverse engineering using fixed points [15]. This is because we only expect the JSLDS approximation to break down if there is a system with nonlinear dynamics that are not well-described by switching between linearizations around fixed points."

---

> > > > ### Comment · Reviewer_DUeH · 2021-08-23
> > > > **Thanks for detailed reply/ remaining issues**
> > > >
> > > > I really appreciate the authors’ detailed reply and additional analyses performed!
> > > >
> > > > I still have some issues/questions though with some of their responses:
> > > >
> > > > Pt.1: I don’t think the Schmidt 2021 study is portrayed correctly. From the examples in that paper it clearly seems the method can learn arbitrary dynamics and is not bound at all to specific fixed point solutions. Rather, in my understanding, the regularization encourages slow components in the dynamics, and the point is more that fixed points can be easily determined *after* training cos it’s piecewise linear.
> > > >
> > > > Pts.14/18: There seems to be a misunderstanding here. Some of the authors’ own text (esp. sect. 4.1) made it appear to me as if the authors were simply forward-iterating the JSLDS equations to find fixed points. For instance this sentence (p.5): “A benefit of JSLDS is that it allows us to find the approximate fixed points of the co-trained RNN by simply running the JSLDS forward in time.” So given this, I wondered why one couldn’t directly run the RNN equations forward in time to find FP? Of course this would miss unstable FP and saddles, and I didn’t quite get it since I thought the e* *are* the approximate FP by means of eq.7? Maybe this (and similar sentences above and below) were just phrased in a somewhat misleading way, or I simply missed a point here.
> > > >
> > > > Pt.16: My point here is that LFADS actually doesn’t learn the underlying dynamics, *in contrast* to the papers I pointed out, but just performs posterior inference. But if the model with which JSLDS is paired does not represent the ‘true’ dynamics after training, this raises the question of what an analysis of its fixed points should reveal about the experimental system? This may be more an issue with LFADS rather than with the authors’ own method. Yet it would have been more interesting to see JSLDS paired with a system that actually *does* map the underlying dynamics.

---

> > > > > ### Author Response · Authors · 2021-08-25
> > > > > **Re: remaining issues**
> > > > >
> > > > > **Pt. 1**: We agree our previous reply may have been unclear/overstated the case regarding the specificity of the fixed point regularization of this method. The more important distinction between our method and this method is the point that our method can be applied to arbitrary (and widely used) RNN architectures.
> > > > >
> > > > > **Pts. 14/18**:  Thank you for the opportunity for further clarification! We think we now understand the cause of the misunderstanding. This will allow us to improve the presentation in the paper.
> > > > >
> > > > > By "approximate fixed points of the co-trained RNN" we meant "using the JSLDS expansion points to approximate the RNN's fixed points/slow points "(as described in Section 3.2). We agree the previous wording was ambiguous.
> > > > >
> > > > > By "running the JSLDS forward in time" we simply meant "computing JSLDS dynamics trajectories for given trials, given the trial inputs", i.e. running forward passes of the JSLDS for given trials. This works because the expansion network depends on the previous JSLDS state, and the next JSLDS state depends on the current expansion point (Section 3.2). So running a JSLDS forward pass given a trial's inputs produces the dynamics trajectory for that trial and the expansion points used to compute them. These expansion points approximate the fixed points used by the co-trained RNN for that trial.
> > > > >
> > > > > We have now clarified these points in the paper and made a more explicit comparison to what is involved in computing the fixed points using the previous method (as mentioned in our previous response).  Please let us know if something is still unclear.
> > > > >
> > > > > **Pt. 16**: This does appear to be a question directed at the LFADS method. We think we understand the question now, but please let us know if we have misinterpreted it. Assuming we understand the question correctly, our understanding is that LFADS does learn a model of the underlying dynamics.  In particular, LFADS learns a generator dynamical system. Given the inferred initial conditions (and in some cases inferred inputs, though not in our example), the generator can be run forward to generate firing rates. This is how the firing rates in Figure 4A are generated and requires learning a good generative model. This generator is the model the JSLDS is used to analyze. We now clarify this aspect of LFADS in our improved experimental discussion based on your previous reply.
> > > > >
> > > > > In general, we agree it will be interesting to see JSLDS applied in other settings in future work.

---

> > > > > > ### Comment · Reviewer_DUeH · 2021-08-28
> > > > > > **Thanks for further clarification**
> > > > > >
> > > > > > I thank the authors for the further clarification.
> > > > > >
> > > > > > Re the last point, it wasn’t immediately clear to me from the text whether the firing rates in Fig. 4 represent a posterior estimate (conditioned on the data through inferred inputs) or were just generated from the initial condition, but I take from the authors’ reply that the latter was the case.

---

> > > > > > > ### Author Response · Authors · 2021-08-28
> > > > > > > **re: Thanks for further clarification**
> > > > > > >
> > > > > > > Yes the latter is the case. This has now been made explicit in the text.

---

> > > > > > > > ### Comment · Reviewer_DUeH · 2021-08-31
> > > > > > > > **update**
> > > > > > > >
> > > > > > > > I thank the authors again for all their clarifications. I'm leaning more toward acceptance now (score updated). I hope I will really find all the promised changes in the final version!

---

### Official Review · Reviewer_DWxE · 2021-07-16

**Rating:** 7
**Confidence:** 4

**Summary:**

This study describes a new interpretable model of neural activity, consisting of a piecewise linear latent dynamical system coupled to a trained RNN. The manuscript outlines intuitions for the model and first applies it to artificial neural networks trained on two cognitive tasks, a working memory task and a context-dependent evidence integration task, showing that their JSLDS model recovers dynamical mechanisms already described in the literature with sets of linear dynamical systems. The authors also propose using the JSLDS model as a replacement for the complex generator RNN of the LFADS model, and apply this idea to neural recordings of a monkey performing a motor control task, showing that a simple linear dynamical system captures most features of this dataset.

**Limitations And Societal Impact:**

yes

**Main Review:**

This work opens a wide range of possibilities by presenting a model with a good balance between interpretability and expressivity, and by showing that it can be applied both to purely in silico studies or fitted to neural recordings.

A weak point of the model in its present state seems to be the expansion network which keeps an unclear division of state-space unlike the idealized version illustrated in figure 1. This seems to lead to an unreasonable number of expansion points according to the other figures.

The model is rather well described, however it could be made clearer in sections 3.2 and 3.3 whether the update matrices of the JSLDS are trained independently of the RNN or whether the derivation operator is backpropagated through from the LDS loss function.

Finally, it would have been interesting to see the JSLDS model used as a lower dimensional model of high-dimensional state-space dynamics, particularly for the neural recordings application, as it is typically done in most LDS studies.

Altogether, the paper exposes convincing applications of a method with a strong potential.

**Time Spent Reviewing:**

4

---

> ### Author Response · Authors · 2021-08-10
> **Reply to reviewer DWxE**
>
> We appreciate the reviewer's very positive feedback regarding the method and results.  We also thank the reviewer for the suggested improvements. We respond to each of these below.
>
> 1. *Regarding an unclear division of state-space by the expansion network*:   We appreciate this comment and now provide a more detailed explanation of the JSLDS fixed point solution in Section 4.1. While there is some variability in the expansion points in the 3-bit memory experiment (Fig. 2E) which causes the points to form clusters instead of single points, this "noise'' is relatively small in the sense that within a cluster, all the expansion points have nearly identical linearizations. We confirmed this by looking at the eigenvalues and eigenvectors. Therefore, these 8 distinct clusters of marginally stable expansion points define what is essentially 8 marginally stable fixed points for each of the 8 possible target output states.
> Upon further investigation, we found that these 8 points are used to dynamically select or ignore the inputs. To see this, we analyzed the left eigenvectors of the Jacobian at each expansion point. These eigenvectors act as selection vectors. When we take the dot product between these selection vectors and the effective input $\frac{\partial F}{\partial u}(e^*, u^*; \theta)(u_t-u^*)$, it reveals visually intuitive patterns that make it clear how the selection vectors are being used to correctly select or ignore the inputs when the system is in a particular state.
> We have now elaborated on these details in Sec. 4.1. We have also added in the 3-bit memory section of the appendix useful plots and a detailed discussion which illustrate the visually intuitive patterns revealed by taking the dot product between the selection vectors and the effective inputs.
> All of this being said, we agree that finding a way to limit the variability in the expansion points for problems with a finite number of fixed points would be an interesting direction for future work. Perhaps an additional penalty in the loss function or a more specific expansion network function could help. We now note this in the conclusion. We have also included in the appendix an extra section that discusses different potential expansion network formulations (such as e.g. possibly taking as inputs the current problem input and/or previous expansion point).
>
> 2.  *On the question related to clarifying the training procedure*: Thank you for this feedback to help us make the presentation of the training procedure more clear. The intention is for everything to be trained together at once. For example, we do not alternate between updating the expansion network parameters $\phi$ or updating the shared RNN and JSLDS parameters $\theta$. Further, the parameters for the JSLDS are the same as the parameters for the RNN (with the addition of the expansion network parameters). For a particular optimization iteration, we compute the loss function in eq. 9, and then update all of the parameters at once through backpropagation. We have added additional wording in Sec. 3.2 and 3.3 to clarify this.
>
> 3. *JSLDS model used as a lower dimensional model of high-dimensional state-space dynamics*: Thank you for pointing out this interesting idea. Currently, the JSLDS must have the same dimensionality as the RNN since its dynamics are parameterized by the Jacobian of the RNN. However,  it would be interesting to try to learn a lower dimensional representation of the RNN dynamics as well. We now mention this as an opportunity for future work in the Discussion.

---

> > ### Comment · Reviewer_DWxE · 2021-08-31
> > **Thanks for the reply**
> >
> > Thank you for your thoughtful reply.

---

### Official Review · Reviewer_cBwm · 2021-07-16

**Rating:** 6
**Confidence:** 4

**Summary:**

This work concerns the challenging problem of improving the understanding of how RNNs perform computations and was inspired by combining ideas from reverse engineering RNNs and SLDS models. The novel SLDS dynamics are governed based on the first-order Taylor series expansion of the co-trained RNN. It is also equipped with an auxiliary function trained to pick out the fixed points of the RNN. Their results are a trained SLDS variant which closely approximates the RNN, an auxiliary function that can produce a fixed point for each point in state-space, and a trained nonlinear RNN whose dynamics have been regularized such that its first-order terms perform the computation, if possible. Moreover, The presented model removes the post-training fixed point optimization, and allows to study the learned dynamics of the SLDS at any point in state-space. It also generalizes SLDS models to continuous manifolds of switching points while sharing parameters across switches.

**Limitations And Societal Impact:**

I think so

**Main Review:**

1) Their contribution (JSLDS) seems genuinely novel.
2) This work is clear and very understandable.
3) The results of this work seem highly important and very interesting.

However, there are a few comments:

- An important part of this work is to approximate the nonlinear RNN by a SLDS with linearizations around fixed points. From Dynamical systems point of view, due to Hartman-Grobman theorem (linearization theorem), the dynamics of a nonlinear system in a domain near a hyperbolic fixed point is qualitatively the same as the dynamics of its linearization near this fixed point. While this is not true in case of non-hyperbolic fixed points. So when there is a non-hyperbolic fixed point, the linearization around the fixed point cannot necessarily describe the dynamics of the nonlinear RNN. But, the authors did not discuss how they could overcome the problem of linearizations around *non-hyperbolic fixed points*, although this needs to be addressed.


- Regarding the sentence " Allows the nonlinear RNN to be approximated by a SLDS, if possible” ( page 2, line 64) :
It is not clear that if their technique fails, this necessarily implies that the nonlinear RNN cannot be approximated by a SLDS.

- Regarding the sentence "The difference between these two solutions shows that the JSLDS co-training procedure can change the fixed points of a nonlinear RNN for a particular task and suggests that it can provide regularization towards simpler solutions for some problems” (page 5, line 183):
There is no justification for the latter have of this claim (the regularization). The authors repeat the claim in their conclusion (283. f) and I’m unsure they actually showed this regularization towards simpler solutions.


**Time Spent Reviewing:**

7 hours

---

> ### Author Response · Authors · 2021-08-10
> **Reply to reviewer cBwm**
>
> We thank the reviewer for the positive feedback. We are glad they appreciate the novelty of the work, the presentation of our idea, and our results. We also appreciate the helpful comments which we respond to below.
>
> 1. *On the Hartman-Grobman theorem*: Thank you for raising this theoretical point. The key is that the higher-order terms in the Taylor approximation are also small. This has been verified in many publications as an empirical fact of line attractors (with a zero eigenvalue) used for evidence accumulation. E.g. Mante, Sussillo et al. 2013 or Maheswaranathan et al. 2019. Additionally, the linearized dynamics do not have to approximate the nonlinear dynamics for all time, merely on the timescale of the input variation.  We now make these points in Section 2.1.
>
> 2. *Regarding the sentence* "*Allows the nonlinear RNN to be approximated by a SLDS, if possible*": We agree this sentence can be made more precise. We have revised this statement as follows: "Allows the nonlinear RNN dynamics to be approximated by switching between local linearizations around fixed points, if possible". This statement better reflects our method since the JSLDS approximation should only fail if the RNN dynamics cannot be described in this way.
>
> 3. *"the regularization towards simpler solutions"*:
>
>     a. We agree our use of "simpler solutions" is unclear. We have revised this sentence to clarify: "The difference between these two solutions shows that for some tasks the JSLDS co-training procedure can change the fixed point solution. This finding along with the improved JSLDS dynamics approximations presented next provides evidence that the JSLDS can act as a regularizer towards solutions better described by switching between linearized dynamics around fixed points.”
>
>     b. We now also note that when initializing the JSLDS co-training procedure with the trained weights of a standard GRU (without the JSLDS regularization), we observe the co-training changes the fixed point solution from the classic solution presented in Fig. 2C to the solution in Fig. 2E. This method also results in the same improved linearized approximations of the dynamics.
>
>     c. Additionally, we have revised the line in the conclusion that you reference to reflect this as well: "In the 3-bit memory task, the JSLDS co-training changed the fixed point structure the co-trained RNN
> used to solve the task compared to the standard GRU solution. We additionally observed improved linearized dynamics approximations with this new solution. This provides evidence JSLDS can regularize a nonlinear RNN towards solutions better described by switching between linearized dynamics around fixed points. A more in-depth study to quantify this potential effect is an interesting line of future work."

---

> > ### Comment · Reviewer_cBwm · 2021-08-17
> > **Problem of non-hyperbolic fixed points**
> >
> > Thanks for the clarification! Most of my concerns are addressed and I like the general idea of the manuscript. However, there is still one concern regarding the problem of non-hyperbolic fixed points. In fact, even if the higher-order terms in the Taylor approximation are  small, mathematically there could be some problems in case of  non-hyperbolic fixed points. Of course, there are many examples which show linearization around non-hyperbolic fixed points has no problem, but it can't definitely be considered as a proof. Moreover, I agree that the linearized dynamics do not have to approximate the nonlinear dynamics for all time. But the problem is that, for non-hyperbolic fixed points, there are cases in which the nonlinear and linearized dynamics are always completely topologically different (i.e. the linearized dynamics cannot approximate the nonlinear dynamics even on the timescale of the input variation). For more clarity, let us concentrate on the reason causing this problem. Also, for simplicity, let us consider continuous-time dynamical systems (as similar results exist for discrete-time DS). Suppose that 0 (the origin) is an equilibrium point for a nonlinear ODE system. Assume further E^s, E^u, and E^c denote stable, unstable and center eigenspaces; and  W^s(0), W^u(0), and W^c(0) denote stable, unstable and center manifolds respectively.  Actually, a key point in the proof of Hartman-Grobman theorem is that when 0 is a hyperbolic equilibrium there is no center manifold and in such a situation tangent spaces E^s and E^u are good approximations for W^s(0) and W^u(0) respectively (particularly since the direction of the vector field on E^s and W^s(0) are the same, as well as the direction of the vector field on E^u and W^u(0)).  But, for the non-hyperbolic equilibrium 0 there is center manifold and, especially, **the direction of the vector field on W^c(0) and E^c can be completely different**. So we can't do a tangent space (E^c) approximation instead of center manifold approximation to investigate the nonlinear dynamics. That is why for non-hyperbolic equilibria, in some cases, **the linearized dynamics cannot approximate the nonlinear dynamics even on the timescale of the input variation**. Especially **they could have different vector field directions at every point**. Certainly, the mentioned problem doesn't happen for all nonlinear systems; but here the authors have considered a class of DS for which such a problem can occur.

---

> > > ### Author Response · Authors · 2021-08-20
> > > **Re: Problem of non-hyperbolic fixed points**
> > >
> > > We appreciate you highlighting this point and the opportunity for further clarification.  We agree in theory there could be general RNN solutions involving non-hyperbolic fixed points where linearizations provide poor approximations of the nonlinear dynamics. We now discuss this potential theoretical limitation in more detail in the discussion section.
> > >
> > > We would like to highlight that our method may provide an important potential advantage that may alleviate this concern for many problems in practice. The goal of the JSLDS co-training is to attempt to learn a good approximation of the nonlinear RNN dynamics by switching between linearizations around fixed points. This is enforced through the regularization terms in the co-training loss function. With proper hyperparameter settings, the co-training should actually be able to bias the nonlinear RNN towards solutions that are well-approximated by this switching between linearizations around fixed points, if such a solution exists.  While it is future work to fully quantify this effect, the improved linearized approximation observed in the 3-bit memory experiment provides evidence this could be the case.

---

> > > > ### Comment · Reviewer_cBwm · 2021-08-21
> > > > **My concerns are addressed well**
> > > >
> > > > Thanks for your reply! I agree with the authors about the mentioned advantage of their method. Also I think pointing out the discussed potential theoretical issue would be enough for this work. Actually, it seems, this issue could be solved using some theoretical methods (in dynamical systems theory). But, it requires some quite detailed investigations potentially as a separate work.

---

### Author Response · Authors · 2021-08-10
**General reply to reviewers**

We would like to thank all four reviewers for the valuable feedback, questions and suggestions. Besides minor edits and clarifications, the major edits are:

1. We have elaborated in much more detail on the JSLDS fixed point solution found for the 3-bit memory experiment.  This also includes a discussion of a new experimental result in which we initialize the JSLDS co-training procedure with the trained parameters of an unconstrained RNN (no JSLDS regularization).  In addition, for the 3-bit memory experiment, we have included useful plots and a discussion in the appendix which provides visual intuition for how the left eigenvectors of the Jacobian for this problem act as selection vectors that select or ignore relevant inputs depending on the current state.

2. For the contextual integration and LFADS experiment, we have included more details both in the experiment sections as well as the appendix for readers that are less familiar with the corresponding papers these experiments are based on.

3. We have included a section in the appendix that discusses different potential expansion network formulations.

---

### Decision · Program_Chairs · 2021-09-27

**Decision:**

Accept (Poster)

**Comment:**

All reviewers agree that the paper proposes an interesting approach to the problem of improving the understanding of RNNs. Although some reviewers have some technical concerns at their first reviews, basically those have been resolved by the authors' responses. Thus, although there are some points that should be modified from the current form, I think we can expect the authors modify the paper in the camera-ready by reflecting the discussion. Based on these, I recommend acceptance (poster) for this paper.